# Atmospheric new particle formation in China

Biwu Chu[1], Veli-Matti Kerminen[1], Federico Bianchi[1, 2], Chao Yan[1], Tuukka Petäjä[1, 3], Markku Kulmala[1, 2]

[1]Institute for Atmospheric and Earth System Research / Physics, Faculty of Science, University of Helsinki, Finland

[2]Aerosol and Haze Laboratory, Beijing Advanced Innovation Center for Soft Matter Science and Engineering, Beijing University of Chemical Technology, Beijing, China

[3]Joint International Research Laboratory of Atmospheric and Earth System Sciences, School of Atmospheric Sciences, Nanjing University, Nanjing 210023, China

*Correspondence to*: Biwu Chu (biwu.chu@helsinki.fi)

**Abstract.** New particle formation (NPF) studies in China were summarized comprehensively in this paper. NPF frequency, formation rate and particle growth rate were closely compared among the observations carried out at different types of sites in different regions of China in different seasons, with the aim of exploring the nucleation and particle growth mechanisms. The interactions between air pollution and NPF are discussed, emphasizing on the properties of NPF under heavy pollution conditions. The current understanding of NPF cannot fully explain the frequent occurrence of NPF at high aerosol loadings in China, and possible reasons for this phenomenon are proposed. The effects of NPF and some aspects of NPF research requiring further investigations are also summarized in this paper.

## 1. Introduction

Atmospheric aerosols have adverse effects on human health and visibility, and cause severe air pollution in many countries (Kaiser, 2005;Cheng et al., 2011;Hand and Malm, 2007;Lelieveld et al., 2015). In addition, aerosol particles influence the Earth's radiation balance due to their direct extinction of light and their capability to serve as cloud condensation nuclei (CCN) or ice nuclei (IN). These influences result in very high uncertainties in predicting the ongoing climate change (IPCC, 2013). In order to understand these effects better, and especially to reduce the uncertainty of evaluating their role in climate change, comprehensive knowledge about the formation and growth of aerosol particles in the atmosphere is required. Atmospheric new particle formation (NPF) is the dominant source of atmospheric aerosol particles on a global scale in terms of number concentrations and has attracted broad attention for couple of decades (Kulmala et al., 2013;Kulmala et al., 2004;Merikanto et al., 2009;Dunne et al., 2016).

Generally, NPF includes the following separate steps: (1) chemical reactions in the gas phase to produce low-volatility vapor(s), (2) cluster formation from gaseous vapors, (3) nucleation or barrierless nucleation, (4) activation of clusters with a second group of vapors to form a critical nucleated particle, and (5) subsequent condensational growth of nucleated particles to detectable sizes or even larger (Kulmala et al., 2014). NPF starts from atmospheric clustering. The key precursors of clusters

have extremely low-volatility, including sulfuric acid (Sipila et al., 2010;Kirkby et al., 2011) and highly oxygenated molecules (HOM) (Bianchi et al., 2016;Tröstl et al., 2016;Kirkby et al., 2016;Ehn et al., 2014). Molecular clusters seem to be continuously generated almost everywhere and all the time (Kulmala et al., 2017). These clusters can be further stabilized by reacting with other gaseous compounds like amines, ammonia and HOM, or through electrostatic interactions in the presence of ions (Kirkby et al., 2016), after which they will grow to larger nanoparticles or will be scavenged by existing surfaces. Therefore, there are two main factors controlling whether NPF will be detected in the atmosphere. One is how fast the clusters grow, while the other is how fast the clusters are scavenged (McMurry and Friedlander, 1979;Kerminen et al., 2001;McMurry et al., 2005;Kuang et al., 2010). Sulfuric acid and organics are the main contributors to aerosol growth. Generally, condensation of sulfuric acid gives an important, sometime dominant, contribution to the initial growth, while organics became more and more important as the particle size is increased (Xiao et al., 2015;Kulmala et al., 2016b). High concentrations of these growth contributors will help the nanoclusters grow to sizes large enough be detected. Meanwhile, pre-existing aerosol particles act as a sink for these precursors, small clusters and newly formed particles, and thereby suppress the occurrence of a NPF event.

Several gas compounds and precursors have been shown to influence NPF under conditions relevant to the atmosphere, such as $SO_2/H_2SO_4$ (Sipila et al., 2010), $NH_3$ (Kirkby et al., 2011;Kürten et al., 2016), amines (Almeida et al., 2013), volatile organic compounds (VOCs)/HOM (Riccobono et al., 2014;Ehn et al., 2014;Bianchi et al., 2016), NOx (Wildt et al., 2014) and iodine species (Sipilä et al., 2016). Meanwhile, many of these compounds are responsible for secondary aerosol formation, which is very pronounced during pollution episodes (Zhang et al., 2015b;Guo et al., 2014). The concentrations of these precursors and preexisting aerosol particles can both be high in polluted cities, especially in the developing countries like China and India, and cause some special features in NPF events compared with cleaner environments, which we cannot explain yet (Kulmala et al., 2017). In China, the rapid economic development and urbanization have led to high emission of various pollutants from coal combustion, motor vehicle exhaust and various industrial emissions, and resulted in highly complex air pollution. Besides high concentrations of fine particles ($PM_{2.5}$, particulate matter with diameter less than 2.5 µm), high concentrations of $SO_2$, $NO_x$, $NH_3$, and VOCs were observed in frequent haze pollution episodes (Liu et al., 2013;Ye et al., 2011;Zou et al., 2015;Wang et al., 2015a). Due to a large proportion of energy supply from coal combustion, the concentration of $SO_2$ was thought to be the highest in the world (Bauduin et al., 2016), with surface concentrations in the range of a few ppb to over 100 ppb in north China (Sun et al., 2009;Li et al., 2007). The emissions and concentrations of $SO_2$ were decreasing in most regions of China during the recent years (Lu et al., 2010;Wang et al., 2015b), but high concentrations (dozens of or over 100 ppb) of $SO_2$ are still being frequently observed during the heating period in winter (Wang et al., 2015a;Zhang et al., 2015a). Unlike $SO_2$, emissions of NOx are also highly related to traffic. NOx emissions in China showed a decreasing trend from 2012 onwards, which appeared later than $SO_2$ (Ronald et al., 2017). Several studies have found that high $PM_{2.5}$ concentrations are strongly associated with the increasing concentrations of NOx (Wang et al., 2013a;He et al., 2014;Ma et al., 2018;Sun et al., 2016). NOx concentrations usually range from a few ppb to dozens of ppb in Chinese cities, while during severe haze pollution

episodes NOx concentration in the city center can be even higher than 300 ppb (He et al., 2014;Sun et al., 2016). For the most important alkaline gas, i.e. NH$_3$, there has been no national scale measurement in China despite its extensive emissions and increasing emission trend (Fu et al., 2015). High concentrations of NH$_3$ (maximum concentration higher than 100 ppb) (Meng et al., 2015b;Wen et al., 2015;Meng et al., 2011;Pan et al., 2012;Meng et al., 2018;Pan et al., 2018) and strong correlations between the peak levels of fine particles and large increases in NH$_3$ concentrations (Liu et al., 2015;Ye et al., 2011) were observed in the North China Plain. Unlike SO$_2$, emissions of NH$_3$ are mainly from non-point sources difficult to control. Emission of VOCs have a similar situation as NH$_3$. The total emissions of VOCs in China were estimated to be still increasing during the recent years (Wei et al., 2011;Wu et al., 2016;Zheng et al., 2018;Sun et al., 2018). Observation data showed that the annual average mass concentration of total non-methane hydrocarbons (NMHCs) was about $10^2$ μg m$^{-3}$, or dozens of ppb in urban and suburban site in Chinese cities, which is much higher than that in North America (Zhang et al., 2017a;von Schneidemesser et al., 2010;Parrish et al., 2009;Zou et al., 2015). HOM can be formed from anthropogenic VOCs (Molteni et al., 2018), although their role on new particle formation is still not clear yet they might play an important role on NPF measured in Chinese Megacities. High concentrations of these gas precursors have resulted in high concentrations of secondary inorganic and organic species in PM$_{2.5}$ during haze formation (Yang et al., 2011;Zhao et al., 2013;Dan et al., 2004;Duan et al., 2005;Wang et al., 2012), but how this cocktail of high concentrations of SO$_2$, NOx, NH$_3$, VOCs and particulate matter (or highly complex air pollution) influence NPF remains highly uncertain.

Atmospheric NPF has been observed globally in almost all kinds of environments (Kulmala et al., 2004;Wang et al., 2017;Kulmala et al., 2016b;Manninen et al., 2010;Nieminen et al., 2018;Kerminen et al., 2018). However, no uniform theory has been found that would explain the occurrence and characteristics of NPF in different atmospheric environments. Generally, NPF was observed less frequently than expected in pristine environments, while more often than theoretical prediction in polluted cities (Kulmala et al., 2017). In this study, we will summarize the NPF studies conducted in China, focusing on the properties of the NPF events in polluted regions and trying to figure out the possible reasons for the frequent occurrence of NPF at high aerosol loadings. Recently, Wang et al. (2017) summarized the techniques, recent advances, current bottlenecks and future directions in studying NPF in China, while this study will provide a more comprehensive summary of the characteristics of NPF and will emphasize the interactions between air quality and NPF.

## 2.   Overview of NPF research in China

Field observation related to atmospheric NPF started around 2004 in China (Wu et al., 2007). After that, observations concerning NPF were carried out in North China Plain (NCP), Yangtze River Delta (YRD), Pearl River Delta (PRD), western China cities like Lanzhou, Xi'an and Urumqi, and coastal cities as well as adjacent seas. NCP, YRD and PRD are the most developed regions in China and they all have a high intensify of population. The air pollution level decreases from NCP to

PRD, or from north to south, among these three regions (Zhang and Cao, 2015). The western China cities like Xi'an and Urumqi suffered by heavy air pollution. Xi'an was reported to have a much higher fine particle concentration than Beijing in NCP (Huang et al., 2014). In 2017, according to the reports of Xi'an Environmental Protection Bureau (http://xaepb.xa.gov.cn/ptl/def/def/index_982_4434_ci_trid_2861812.html, last access: 12 October 2018), Xinjiang

Department of Environmental Protection (http://www.xjepb.gov.cn/xjepb/resource/cms/article/2012/268650/2017.pdf, last access: 12 October 2018) and Beijing Municipal Environmental Protection Bureau (http://www.bjepb.gov.cn/bjhrb/resource/cms/2018/05/2018051614522475279.pdf, last access: 12 October 2018), the annual average $PM_{2.5}$ concentrations were 73 µg/m$^3$ and 70 µg/m$^3$ in Xi'an and Urumqi, respectively, which were higher than that of Beijing (58 µg/m$^3$).

A map of observation stations involving NPF study in China is shown in Figure 1. These observation sites include urban and suburban sites like Beijing, Shanghai, Nanjing, Guangzhou etc., regional and rural sites like Shangdianzi, Yufa, SORPES (Nanjing University), Backgarden, Kaiping, etc., and mountain sites like Waliguan, Tai, Heng, Huang etc., providing information on aerosol size distribution in different environments. Besides routine observations, comprehensive campaigns like PRIDE-PRD2004, CAREBeijing-2006, and CAREBeijing-2008 were also carried out for better understanding of NPF

and aerosol pollution in representative region and period in China. Long-term observations on NPF are relatively rare in China, and only a few studies reported NPF observations covering longer than a one-year period (Wu et al., 2007;Kivekas et al., 2009;Yao et al., 2010;Wu et al., 2011;Shen et al., 2011;Qi et al., 2015;Peng et al., 2017). The relative short-period observations may not represent varying atmospheric conditions, and therefore, the applicability of these observation results may be limited to specific conditions.

Up to now, about 100 papers from about 20 groups have been published related to NPF in China. Most of these studies focused on the characterization of NPF events, such as the properties and time evolution of the particle size distribution, particle formation and growth rate, condensation sink, etc. Some of them also studied the favorable conditions for NPF, including the influences of relative humidity (RH), temperature, wind speed and direction, air mass origin etc. Few of these studies investigated NPF mechanisms involving the nucleation participants, the growth contributors and the scavenging process by

preexisting aerosols, while some others investigated various effects of NPF, especially the contribution of NPF to atmospheric CCN.

The aerosol size distribution and its time evolution provide the basic information for studying NPF. Many studies about NPF in China measured only particles larger than 10 nm, while a few studies also measured particles with diameters in the range of 3-10 nm. In recent years, an increasing number of studies were carried out with measurements of sub-3 nanoparticles

(Xiao et al., 2015;Cai et al., 2017;Cai and Jiang, 2017;Jayaratne et al., 2017;Dai et al., 2017;Lv et al., 2018;Yao et al., 2018), using Particle Size Magnifiers (PSM), neutral cluster and air ion spectrometers (NAIS) or diethylene glycol scanning mobility particle spectrometers (DEG-SMPS). As for gas phase precursors, direct measurements of $H_2SO_4$ were carried out with

atmospheric pressure-ion drift-chemical ionization mass spectrometer (AP-ID-CIMS) in a few studies (Yue et al., 2010;Zheng et al., 2011;Wang et al., 2011), while other studies usually estimated $H_2SO_4$ concentrations using different proxies related to $SO_2$, radiation, $O_3$, and relative humidity (RH). Amines and ammonia are crucial in NPF since they are able to stabilize sulfuric acid clusters by forming acid-base complexes, yet there are very little NPF measurement results related to these compounds in

China (Zheng et al., 2015b;Yao et al., 2016;Yao et al., 2018). Measurement results on natural ions and neutral compounds/clusters, including both $H_2SO_4$ and HOM, obtained using an atmospheric pressure interface time-of-flight mass spectrometer (APi-TOF-MS) and a nitrate-based chemical ionization–APi-TOF-MS (CI-APi-TOF-MS), were recently reported by Yao et al. (2018). Research on these relevant gaseous compounds like HOM, or on air ions, is still very limited in China yet. Comprehensive, long-term and high quality relevant measurements are required for a better understanding about

the nucleation and growth mechanisms of nanoparticles in China.

## 3.   Characterization of NPF events in China

There are a few basic parameters to characterize NPF events, which are listed in Table 1. Most of the NPF researches in China calculated these parameters, as listed in Table 2. In the following chapters, we will summarize and discuss the measurement results of the frequency of NPF events, new particle formation rate (FR), particle growth rate (GR) and the related

concentrations and source rate of condensation vapors. Although there were differences in calculating these parameters by different groups, we will not discuss much about the methodology since the main purpose of this paper is to provide an overview of the characteristics of NPF in China.

### 3.1 NPF frequency

The primary information in studying atmospheric NPF is whether it is taking place or not, i.e. to identify NPF events.

Unfortunately, there is no unique mathematical criterion or definition for a NPF event. Dal Maso et al. (2005) suggested criteria for justifying a NPF event: a distinctly new mode of particles start in the nucleation mode size range, prevails over a time span of hours, and shows signs of growth. The particle growth is important for separating a NPF event from particles associated with local emission sources like traffic, especially when the particle size detection limit of the instruments is not low. In addition to NPF event days, the days with an absence of particles in the nucleation mode size range are called non-event days.

However, some days are not easy to be classified as either events or non-events, so they usually classified as undefined days. Most NPF studies in China used similar methods, but certainly subjective biases existed. A challenge that exists for identifying NPF is the interference of primarily emitted particles from local combustion sources near the observation site. For example, the formation and rapid growth of vehicular particles during the initial 1–2 s of exhaust cooling and dilution processes frequently lead to a nucleation mode at 10–20 nm (Vu et al., 2015;Lee et al., 2015). Spikes of particle number concentration

associated with combustion emissions were observed in many NPF studies (Liu et al., 2014;Wang et al., 2014a;Peng et al., 2017;Zhu et al., 2017), but these spikes usually had some different characteristics from those of the NPF events (Wang et al., 2014a). The particle size (Hofman et al., 2016), the ratio of number concentrations of in the nucleation mode particles to those of fine particles (Peng et al., 2017;Jung et al., 2013), the time of duration of NPF events (Zhu et al., 2017), and the correlation of the particle number concentration with other gaseous pollutants concentrations and meteorology conditions (Wang et al., 2014a) were used to identify the contribution of primary emission in the burst of particle number concentration. However, there are still uncertainties in distinguishing the new-particle signal from the mixed signals of newly-formed particles and freshly-emitted particles from combustion, especially when NPF measurements were carried out with a particle size detection limit larger than 10 nm. There is a possibility that the growth of the vehicular emission of sub-10 nm particles may look like a NPF event and therefore overestimate the NPF frequency. A recently observation found a notable presence of traffic-originated nanocluster aerosol particles in the size range of 1.3–3.0 nm in urban air (Rönkkö et al., 2017), which might raise new questions about the sources of nanocluster aerosol particles in semi-urban roadside environment. In this study, as mentioned earlier, we will not pursue the details of the justifying methods, but focus on the statistic results of the measurements.

NPF events were observed with quite different frequencies ranging from less than 10% to more than 50% in different environments and different seasons. In Figure 2, we summarize the reported NPF event frequencies in China according to the season, observation site type and region, but ignoring observations of too short period like less than one month. Generally, low frequencies were observed in remote clean environments like on marginal seas (Liu et al., 2014), while there were no significant differences among urban, suburban and rural or regional sites. Although higher NPF event frequencies were sometimes observed in an urban site compared with a rural site in the same region (Yue et al., 2013), NPF was usually found to be a regional phenomenon in China. For example, NPF in the Beijing urban area always coincided with NPF at a regional site 120 km away (Wang et al., 2013c). Shen et al. (2018) observed regional NPF in NCP with a horizontal extent larger than 500 km and found that large-scale regional NPF was favored by a fast transport of northwesterly air masses. Despite the similar frequencies, much higher FR (by 220%) and GR (by 50%) were observed in the Beijing urban site than in the corresponding regional background site (Yue et al., 2009;Wang et al., 2013c). The corresponding values of source rate of condensation vapors (Q), condensation vapor concentration (Ccv), and condensation sink (CS) were also larger at Beijing than those at the regional site Yufa by 40%, 40%, and 60%, respectively (Yue et al., 2009). These results indicated that the higher pollution level in Chinese cities usually resulted in stronger NPF events compare to rural areas. As for different regions, there seemed to be no significant differences in the NPF event frequency between NCP, YRD and western China cities. PRD had a relatively lower NPF frequency compared with these three regions, but the difference was not statistically significant. Despite different pollution conditions in different regions of China, there is a lack of long-term NPF observations, which limits our knowledge about the relationship between the level of air pollution and the occurrence of NPF.Air pollutants and meteorological features are usually studied together with nanoparticles and their precursors. By comparing the pollution character between NPF events

and other days, the primary factors affecting NPF events might be identified. Cai et al. (2017) found that the Fuchs surface area (which is a representative parameter of coagulation scavenging based on kinetic theory and is proportional to CS) fundamentally determined the occurrence of NPF events in Beijing. The Fuchs surface area had a good correlation with the $PM_{2.5}$ mass concentration, and no NPF event was observed when the daily mean $PM_{2.5}$ concentration was higher than 43 μg

$m^{-3}$ in the winter of 2015 in Beijing (Jayaratne et al., 2017). However, in some cases, the CS or the average coagulation sinks during NPF events were not significantly lower compare to other times when new particles were not formed, indicating that other factors, such as the precursor vapors and photochemical activity, might also play an important role in driving NPF (Gong et al., 2010). Besides the condensation sink, NPF events seemed not to be very sensitive to the concentration levels of common gas pollutants in China, such as $O_3$, $SO_2$, and $NO_2$ (Zhu et al., 2013;An et al., 2015). It was observed that $SO_2$ concentrations

were lower during the NPF event days than during non-event days in NCP (Herrmann et al., 2014) and Taiwan (Young et al., 2013a) as well as during autumn and winter in YRD (Qi et al., 2015), whereas higher $SO_2$ concentrations on NPF days were only observed during spring and summer in YRD (Qi et al., 2015;Yu et al., 2016), during autumn in PRD (Gong et al., 2010), and at mountain sites (Zhang et al., 2017c). Meanwhile, based on the empirical parameter developed to judge whether NPF will occur or not, the exponent of $SO_2$ in this empirical parameter was quite small, indicating there is usually enough $SO_2$ for

NPF to occur under heavily polluted conditions (Herrmann et al., 2014). Similar results for sulfuric acid were reported and it was found that sulfuric acid concentrations were not significantly higher (even lower, sometimes) on NPF days compared with non-event days (Qi et al., 2015;Xiao et al., 2015;Cai et al., 2017). Overall, the previous results seem to suggest that $SO_2$ was not a limiting factor for NPF in China, and similar conclusion might also be made for sulfuric acid. However, higher $SO_2$ concentrations could increase the probability of occurrence of NPF events at a mountaintop site (Lv et al., 2018). Besides,

NPF might have different patterns in environment with abundant $SO_2$ or not. Stronger nucleation but weak growth of particles were observed with high concentrations of $SO_2$ in polluted air masses characteristic of urban (heavy traffic emission) or power-plant plumes, in spite of similar CS with lower concentrations of $SO_2$ (Gao et al., 2009;Yue et al., 2010).

        NPF event frequencies were different between the different seasons. In north China, spring is usually the season with the highest frequency of NPF events, which is probably due to the typically low CS, relatively high solar radiation intensity, and

low temperature and RH (Shen et al., 2011;Wu et al., 2007). In the NCP of China, many studies observed that summer had the lowest NPF event frequency although the condensable vapor concentration was the highest during summer months due to enhanced photochemical process (Shen et al., 2011;Wu et al., 2007;Yue et al., 2009). The lowest frequency of NPF events during summertime in NCP might be related to the high temperatures and RH, together with the stagnant and polluted air masses which could cause a high CS (Wu et al., 2007). In the YRD region, high NPF event frequencies were observed in spring

and summer, although the temperature and RH were high in summer (Zhu et al., 2013;Qi et al., 2015). A low temperature favors NPF (Zhu et al., 2013), but according to the our summary, as shown in Figure 2, low NPF event frequencies were usually observed in winter, which might be due to the weak solar radiation as well as typically high pollution levels at that

time of the year. In spite of an increasing number of aerosol size distribution measurements in China, atmospheric NPF observations that cover the full annual cycle are still quite limited. Meanwhile, the main reason for the different NPF event frequencies in different seasons is still uncertain because many factors influencing NPF, such as the radiation intensity, temperature, relative humidity, wind properties, biogenic activity and anthropogenic emissions, tend to be changed

simultaneously.

     The NPF event frequency can also be quite different in air masses from different directions (Wu et al., 2007). Higher NPF event frequencies were usually observed within relatively clean air masses having a low CS (Zhu et al., 2013;An et al., 2015;Jayaratne et al., 2017;Peng et al., 2017). However, in some cases, NPF events also occurred in polluted air masses. For example, during the summer in Beijing, NPF was observed under low wind speed conditions and this phenomenon usually

coincided with a wind direction change from north to south, where the air is more polluted (Zhang et al., 2011). Similarly, in Hong Kong NPF was usually observed when air masses originated from the northwest to northeast directions (Guo et al., 2012). At the summit of Mt. Tai, continental air mass passing through more polluted areas also favored NPF (Lv et al., 2018). Consecutive NPF events were observed in the presence of strong biomass burning plume at a downwind rural site in PRD (Wang et al., 2013e). Meanwhile, compared to the NPF events taking place in clean air masses, the FR seemed to be lower and

the GR seemed to be higher in the NPF events taking place in a polluted air mass plume (Qi et al., 2015). An observation in North China Plain reported that when the air mass was transported from the polluted south area, the average $PM_{10}$ (PM with diameter less than 10 µm) concentrations in NPF event days were higher than during the non-event days (Shen et al., 2011). In addition, air masses from polluted north China favored the occurrence of regional NPF, while clean air mass from east usually caused local NPF in Nanjing in YRD region (Dai et al., 2017). These results highlighted the complex relationship

between air pollution and NPF. Many factors, including pre-existing aerosols, organic pollutants and $SO_2$, are connected each other due to their similar emission sources, so it is not easy to extract the influence of one factor on NPF. Furthermore, since environments are complex and diverse, some other factors, such as the concentration of OH radicals and topography, can also be important to NPF and therefor deserve further investigation in both field observations and controlled experiments.

## 3.2 Formation rate

Due to the lack of measurements down to particle diameters of about 1.5 nm, most atmospheric nucleation rates were inferred indirectly only by measuring the particle formation rate at some larger size in most of the NPF studies in China. The FR at larger sizes (the "apparent" particle formation rate) can be related to the FR of critical clusters (the "real" nucleation rate) by Kerminen–Kulmala equation and its revised version (Lehtinen et al., 2007;Kerminen and Kulmala, 2002), but the nuclei GR and coagulational scavenging rate (CoagS or CS) are needed. Besides, the assumed coagulation sticking probability of 1 for

molecular clusters with pre-existing particles in their collision and the unclear GR of sub-3 nm particles might result in errors

in the derivation of FR (Kulmala et al., 2017).We did not convert the "apparent" particle formation rate into "real" nucleation rate, but summarized FR calculated at different particle sizes in this study (Figure 3). The observed FR ranged from less than 0.1 cm$^{-3}$ s$^{-1}$ at particle sizes larger than 10 nm to about 10$^3$ cm$^{-3}$ s$^{-1}$ at particle sizes below 2 nm. At a certain particle size, the FR could still differ by 2 orders of magnitude due to the different environmental conditions. For example, many studies reported the FR of 3 nm particles ranging from less than 1 to several tens of cm$^{-3}$ s$^{-1}$.

Due to the wide range of FR under different environmental conditions, it is not easy to determine differences in FR between different site types, regions or seasons. In principle, a higher CS causes a more rapid scavenging of clusters and small particles, resulting in lower FR (Zhu et al., 2014;Man et al., 2015). According to the equation developed by Herrmann et al. (2014) based on observation date in YRD region, FR is also inversely proportional to the CS. However, when NPF was studied in an urban site and a nearby regional site at the same time, FR was usually higher at the urban site in spite of the higher CS, indicating much more abundant precursors for NPF in the polluted urban environment (Wang et al., 2013c). As for NPF at a same observation site but in different seasons, the highest FR was observed in summer in NCP (Shen et al., 2011) and in spring in YRD (Qi et al., 2015).

Although the nucleation mechanism in different environmental conditions remains unknown according to current knowledge, neutral clusters of sulfuric acid, stabilized with additional vapours such as ammonia, amine, HOM should play a key role in NPF (Kulmala et al., 2014;Kulmala et al., 2013). A positive relationship between nucleation rate and the sulfuric acid concentration (or $H_2SO_4$ proxy) was observed in many NPF studies in China, although nucleation rates were rarely calculated using measurements of particles in the size range of 1-3 nm. The fitted exponent between FR and sulfuric acid concentration ranged from 0.65 to 2.4 (Cai et al., 2017;Xiao et al., 2015;Dai et al., 2017), while sometimes even higher values between 2.5 and 7 were found (Wang et al., 2011). These exponents were observed to increase with an increasing CS in Beijing (Wang et al., 2011). Besides sulfuric acid, also organics, $NH_3$ and amines were found to be important in atmospheric particle nucleation (Wang et al., 2015c). As we mentioned earlier, although CS was much higher at urban sites, the FR was usually higher at corresponding regional sites (Wang et al., 2013c). Meanwhile, $SO_2$ is a regional pollutant and its concentrations were similar between regional sites and city area (He et al., 2014;Ma et al., 2018). These features indicate important roles of other gas precursors in NPF in the air pollution complex of China. In fact, some observations showed that the correlation between FR and $NH_3$ was better than that between FR and $H_2SO_4$ (Xiao et al., 2015). According to the national ammonia observation network, the overall average concentration of ammonia in China is much higher than the values observed in the U.S. The seasonal maximum $NH_3$ concentrations were observed in the summer and the most abundant concentrations of $NH_3$ were observed in the NCP region in China (Pan et al., 2018). Compare to $NH_3$, the amine measurements are more sparse (Zheng et al., 2015b;Yao et al., 2016), and direct information on amine emissions is currently not available but these emissions have to be estimated by assuming a fixed ratio or source-dependent ratios of amines to total ammonia emissions in China (Mao et al., 2018). Dai et al.(2017) proposed that plumes containing high concentrations of ammonia, amines or HOM produced from their

observed VOCs led to strong local NPF events. The observations made at the SORPES station in YRD indicated that HOM played an essential role in the initial condensational growth of newly formed clusters (Huang et al., 2016;Ding et al., 2016;Qi et al., 2018). Recently, Yao et al. (2018) reported a long-term continuous observations for NPF in urban Shanghai and observed one to two orders of magnitude higher FR than typical values in the clean atmosphere. These observed FR were far higher than

those derived from $H_2SO_4$–$H_2O$ or $H_2SO_4$–$NH_3$–$H_2O$ mechanisms but close to those observed in the $H_2SO_4$–DMA–$H_2O$ laboratory experiments, and coincided with sulfuric acid clusters and sulfuric acid–dimethylamine (DMA) clusters. These results suggested that $H_2SO_4$–DMA–$H_2O$ nucleation played important roles in the NPF in Chinese megacities. Up to now, there is still quite limited investigations for the relation between FR and organics, $NH_3$ and amines in China and it is certainly crucial for a better understanding of NPF in polluted area. Ion-induced of pure organic nucleation was proposed to be important

according to chamber experiments (Kirkby et al., 2016), but seems to have a minor role in the polluted environment in China (Herrmann et al., 2014;Xiao et al., 2015;Yao et al., 2018). This is understandable because the ion production rate is usually much lower than FR in China.

## 3.3 Growth rate

Growth of nano-particles is crucial for NPF. The GR determines the size that new particles can grow to before being scavenged,

i.e. a higher GR results in a larger particle diameter (Zhu et al., 2014;Man et al., 2015). There are several methods to calculate GR from the time variations of particle size distributions, such as the appearance time method (Kulmala et al., 2013) and mode fitting method (Kulmala et al., 2012), or by solving the general aerosol dynamics equation (Pichelstorfer et al., 2018).

Regardless of the possible difference caused by using different calculation methods, GRs reported in China varied a lot from an urban area to a rural region and from spring to winter, ranging from a few nm h$^{-1}$ to more than 20 nm h$^{-1}$ (Table 2).

Generally, GRs in urban sites were found to be higher than in their regional sites, as shown in Figure 4(a) (Wang et al., 2013c) , which is also summarized by Kerminen et al. (2018). This is probably caused by the more abundant condensing vapors in polluted cities, although there are limited data on sulfuric acid and low-volatile organic vapor concentrations in China. No significant differences were found among the observations carried out in different regions in China in spite of the different pollution levels. Lower GR was observed at a mountain site compare to that in an urban area (Wang et al., 2014b), but the GR

of large-size particles in mountain site could be as high as about 10 nm h$^{-1}$ (Nie et al., 2014;Wang et al., 2014b). For GRs in different seasons, higher GRs were observed in summer than other seasons, indicating higher concentrations of condensable vapors, which may be related to strong photochemical and biological activities, as shown in Figure 4(a, b) (Zhu et al., 2013;Shen et al., 2011;Qi et al., 2015). GR is also dependent on the concerned size of the particles, larger particles usually having a larger GR (Xiao et al., 2015), which could also be inferred from the data summarized in Figure 4(c).

Sulfuric acid and organic vapors with low volatility were thought to be the main contributors to the growth of particles

formed by NPF. Generally, sulfuric acid was thought to be the dominant contributor for the growth of newly formed particles, but became less and less prominent for the growth of larger particles (Xiao et al., 2015). For example, Xiao et al. (2015) and Yao et al. (2018) calculated and estimated that sulfuric acid was enough to explain the observed growth for particles smaller than 3 nm, but was insufficient to explain the observed growth rates of large particles. They further calculated the relative

contribution of sulfuric acid to the particle growth in different particle size ranges. As shown in Figure 4(c), these calculated contributions were 39% and 29% for the size ranges of 2.39-7 and 7-20 nm, respectively, in urban Shanghai (Xiao et al., 2015), 3% to 14% for the size range of 7-30 nm in urban Beijing (Wang et al., 2015c), about 26 % for the size range of 5.5-25 nm in suburban Hong Kong (Guo et al., 2012), and about 29% during the Beijing Summer Olympic period (Gao et al., 2012). Some studies reported that $H_2SO_4$ had a negligible contribution to the growth of particles larger than 10 nm (Meng et al., 2015a;Liu

et al., 2014). The particle shrinkage (reversal in growth of particles size) was reported in a few studies in China. The particle shrinkage could be due to measuring particles present in different air masses during different times of the day, or the evaporation of water and/or semi-volatile species in the particles. If the air masses did not vary significantly, a similar shrinkage rate to the growth rate in the NPF events might indicate a notable fraction of semi-volatile species contributed to the growth (Young et al., 2013b;Yao et al., 2010), which is consistent with organics being the main contributor to the large-size particle

growth. Yu et al. (2016) estimated that a high concentration of extremely low volatility organic compounds was the key factor leading to a maximum in GR for very small particles (1.4-3 nm) in urban Nanjing. Although the existence of local maxima in GR in the sub-3 nm size range is highly sensitive to uncertainties in particle size distributions, the results highlighted that detailed investigations for the mechanisms of the initial growth steps of atmospheric NPF are needed (Yu et al., 2016). On the other hand, Yue et al. (2010) proposed a dominant role of sulfuric acid in the growth of new particles in sulfur-rich NPF events.

A model simulation study about NPF in Beijing also supported that only small fraction of organics contributed to the growth of new particles, and these organics were mainly $O_3$ initiated (Wang et al., 2013b). Besides sulfuric acid and organics, some studies reported a two-stage growth of new particles in China, in which sulfuric acid and organics contributed to the first-stage growth in daytime while $NH_4NO_3$ and organics possibly contributed to the second-stage growth at nighttime (Zhu et al., 2014;Man et al., 2015;Liu et al., 2014). Tao et al. (2016) observed higher levels of aminiums in particles with relative smaller

sizes, and suggested that the heterogeneous uptake of amines by acid-base reactions could effectively contribute to the particle growth during NPF events. However, they only measured the particle chemical composition with a lowest cutoff size of 56 nm, which may not be directly related to NPF. In fact, measuring the chemical composition of nucleation-mode particles is still quite challenging all over the world. To summarize, most studies observed a slow GR for newly formed particles, with $H_2SO_4$ as the dominate contributor, while more other species, such as organics, would contribute to the particle growth as the

particle grow to bigger size and also result in higher GR.

## 4. NPF under heavy air pollution

The heavy air pollution makes China quite a different environment for NPF compared with western countries (Wang et al., 2017;Kulmala et al., 2017;Yu et al., 2017). Generally, concentrations of particles and condensable vapors in Chinese cities and regional background area are much higher in China than that in North America or Europe (Shen et al., 2016b;Wang et al., 2013c;Gao et al., 2009). The CS and small molecular cluster and particle (1–3 nm) concentrations are about an order of magnitude higher in China compared with European cities (Kulmala, 2015;Kontkanen et al., 2017). The occurrence frequencies of NPF events in high aerosol-loading environments of China were higher than those in low aerosol-loading environments (Peng et al., 2014). Meanwhile, the observed FR was much higher, and the GR was also higher (but to a less great extent relative to FR) for NPF in China than that at rural/urban sites in western countries (Shen et al., 2016b). As pointed out by Cai et al. (2017), previous FR calculation may still underestimate the real nucleation rate due to underestimation or omission of coagulation among particles in the nucleation mode with strong nucleation in China.

The influence of heavy air pollution on NPF might be identified by studying NPF in periods with short-term strong air pollution control. Shen et al. (2016c) investigated NPF during the Olympics in 2008 and during APEC meeting in 2015 in Beijing. They found that a higher NPF event frequency coincided with the improved air quality during these important events associated with temporary intense air pollution control actions compared to similar time of the year during 2010-2013. In spite of more frequent NPF events (Yue et al., 2010;Zhang et al., 2011), the strength of NPF decreased during these periods with temporary intense air pollution control actions, characterized with lower FR and GR values (Shen et al., 2016c). Due to the decreasing strength of NPF and also the reduction of primary emission source of fine particles, the number concentration of particles decreased in spite of increased frequency of NPF. The mean number and volume concentrations of particles decreased by 41% and 35%, respectively, in August 2008 during the Beijing Olympic compared with 2004-2007 (Wang et al., 2013d). However, these temporary intense air pollution control actions had a much smaller influence on Aitken mode particles than on accumulation mode particles, according to the observations carried out during the APEC meeting in 2015 in Beijing (Du et al., 2017).

NPF was observed more often in high aerosol-loading environment of China than we would expect based on the current understanding of nucleation and particle growth (Peng et al., 2014;Kulmala et al., 2017). The ratio of particle scavenging loss rate over condensational growth rate, which is proportional to the ratio of CS to GR, was used as a criterion to predict the occurrence of NPF events (McMurry et al., 2005;Kuang et al., 2010). With much higher CS values in China than that at European and American sites, the difference of GR was not very obvious at the same site types between China and other countries (Peng et al., 2014). It turned out that NPF occurred frequently in megacities in China when the ratio of CS ($10^{-4}$ s$^{-1}$) to GR (nm h$^{-1}$) was above 200, whereas it only occurred when this same ratio was less than 50 under clean and moderately-polluted conditions (Kulmala et al., 2017). As shown in Figure 5, most of the observation data reported ratios of CS ($10^{-4}$ s$^{-1}$)

to GR (nm h$^{-1}$) between 200 and 500, while a few less than 200 but always higher than 50. More importantly, many studies reported NPF to take place with this ratio higher than 500 at urban and suburban sites. Such NPF events were able to take place in all regions in China (NCP, YRD, PRD and Western China), and during both winter and summer seasons. There are several possible reasons for the higher threshold ratio of CS to GR in highly polluted environment, including the overestimation of particle losses due to assuming a coagulation sticking probability of 1, the underestimation of GR in the sub-3 size range, and also unrevealed nucleation and growth mechanism relevant to a polluted atmosphere (Kulmala et al., 2017;Yu et al., 2017).

NPF mainly occurred when the $PM_{2.5}$ concentration (CS) and gas pollutant concentrations, such as $NO_2$, CO and $SO_2$, were both low (Wu et al., 2007;Dai et al., 2017;Yu et al., 2016). These gas pollutants were mainly from primary combustion emission (De Gouw and Jimenez, 2009), whereas $PM_{2.5}$, the main cause of haze, originated from both primary emission and secondary formation and the latter was thought to dominate during haze events in China (Yang et al., 2011;Zhao et al., 2013;Dan et al., 2004;Duan et al., 2005;Wang et al., 2012). NPF was found to concentrate on days with low RH in previous NPF studies in Beijing (Wu et al., 2007;Yue et al., 2009). The possible reason for this would be that photochemical reactions are faster on sunny days with strong solar radiation and low RH. On the contrary, haze usually occurs at high RH when multi-phase processes would contribute more to the aerosol mass (Sun et al., 2010;He et al., 2014;Zheng et al., 2015a;Cheng et al., 2016;Liu et al., 2017) and the hygroscopic aerosols would contribute more to light extinction compared with low-RH conditions (Shi et al., 2014;Shen et al., 2015). NPF and haze are either purely secondary processes or dominated by secondary pollution processes, so there might be some common properties or internal relations between them. With this in mind, besides the possible inaccurate estimation for the GR and CS as pointed out and estimated by Kulmala et al. (2017) and further discussed in detail by Yu et al. (2017), several other possible reasons might also be related to the frequent occurrence of NPF under heavy air pollution in China.

Secondary aerosols, including sulfate and organic aerosols, are still underestimated in current air quality models (Xiao et al., 2015;Chen et al., 2016;Hodzic et al., 2010), indicating unknown chemical and physical processes important for secondary aerosol formation (Kulmala et al., 2014). These processes might create oxidants or change the surface properties of aerosols, and thereby limit their ability to take up condensable vapors and cause more frequent NPF (Kulmala, 2015). The effects of a high percentage of inorganic aerosol particles on the effectiveness of CS is unknown and may need to be investigated in laboratory experiments. NPF was observed during dust episodes in China, and both FR and GR were enhanced under dust conditions, indicating photo-induced, dust surface-mediated reactions might be important by producing condensable vapors for NPF (Nie et al., 2014;Xie et al., 2015;Kulmala et al., 2017). Heterogeneous photochemical processes inducing new particle formation and growth might happen in the real atmosphere and need to be further investigated. In addition to these, high concentrations of sulfuric acid ($10^6$ molecules cm$^{-3}$) were observed at nighttime, indicative of non-photochemical OH sources (Zheng et al., 2011). The contribution of oxidation of $SO_2$ by Criegee radicals (Welz et al., 2012;Mauldin et al., 2012), which are generated in reactions of alkenes and ozone, and other possible surface-mediated reactions that lead to the formation of

nighttime sulfuric acid under complex air pollution conditions in China, need to be figured out as well.

The NPF leads directly to a burst of small nanoparticles and increases the particle number concentration prominently. While NPF usually tends to occur in clean days with low CS, particle number concentrations are usually much higher on NPF event days than on non-event days (Shen et al., 2016a;An et al., 2015). Kulmala et al. (2016a) studied separately nucleation, Aitken and accumulation mode particle number concentrations in Nanjing in YRD regions of China, and estimated that the majority of the particles were of secondary origin in all the modes. NPF was found to be an important influential factor on atmospheric aerosol number size distribution from remote mountains to polluted cities (Du et al., 2012;Shen et al., 2016a;Zhang et al., 2017c). NPF also changes the surface and volume size distribution. An et al. (2015) observed that NPF events had a large effect on Aitken- and nuclei-mode particle surface and volume concentrations, while having limited contributions to accumulation- and coarse-mode particles. NPF was observed to increase the proportions of $NH_4^+$, $SO_4^{2-}$, $NO_3^-$, $K^+$ and $Mg^{2+}$ in nucleation- and Aitken-mode particles compared with those in the total aerosol. Zheng et al. (2011) found that the calculated condensation rate of $H_2SO_4$ correlated with the Aitken-mode sulfate mass concentration but not with the accumulation-mode sulfate mass concentration.

With high concentrations of condensable vapors, newly-formed particles have a potential to grow quickly, which results in an increase of PM volume or mass concentrations. In an episode with consecutive NPF events in the presence of strong biomass burning in PRD, the aerosol volume concentration increased by 6.1 mm$^3$ cm$^{-3}$ in volume mass concentration per day or about 10 μg m$^{-3}$ per day in mass concentration, with organics and sulfate accounting for 42% and 35%, respectively, of the particle mass concentration (Wang et al., 2013e). Furthermore, it was estimated that primary emissions and secondary formation provided 28 and 72% of particle number concentration and 21 and 79% of mass concentration, respectively. Similarly, Shen et al. (2011) observed that about 20% of the NPF events led to a measurable increase in the particle mass concentration, with an average growth rate of about 4.9 μg m$^{-3}$ h$^{-1}$ for $PM_1$ (PM with diameter less than 1 μm) during the period of the mass concentration increase. Guo et al. (2014) reported a case with NPF followed by a continuous growth and appearance of haze pollution in Beijing, and proposed that the efficient aerosol nucleation and growth led to severe $PM_{2.5}$ development.

In summary, NPF was found to be the main source of the particle number concentration in the atmosphere, being able to dramatically increase particle number concentrations in a relatively short time. NPF and subsequent particle growth seem also to have a noticeable contribution to the volume and mass concentration of nucleation- and Aitken-mode particles. Although secondary formation of $PM_{2.5}$ mass is the main cause of haze compared with primary particle emissions, the accumulation of this secondary aerosol mass usually occurs over several days following NPF. The contribution of NPF to haze formation is still an open question.

## 5. Significance and future research directions for NPF study

The effects of NPF on air pollution and human health are crucial but highly uncertain. As we mentioned above, the effect or contribution of NPF to haze formation is still an open question. Answering this question might be difficult using only field observations, so new laboratory experiments and model simulations may need to be designed. In addition, interactions between NPF, pollution and meteorological conditions should be studied further. Heavy pollution could have significant feedbacks to meteorological conditions in China. For a case study in YRD, it was calculated that air pollution resulted in a decrease in the solar radiation intensity by more than 70 %, decrease in the sensible heat by more than 85 % and temperature drop by almost 10 K (Ding et al., 2013). These effects resulted in a decrease of the boundary layer height, which further increased PM concentrations, forming a feedback loop (Petaja et al., 2016). On the other hand, NPF occurring in a free troposphere may have a major impact on the marine boundary layer particle concentrations due to the subsidence (Clarke et al., 1998;Lin et al., 2007). When the aerosol loading was high, the distance of NPF peak above the planetary boundary layer became larger (Quan et al., 2017). These interactions would be also crucial for predicting NPF and air quality and for identifying the contribution of NPF to air pollution. Rather than ground observations, multi-dimensional measurements may need to be carried out in order to understand the atmospheric process up to the free troposphere. Compared with the effect of NPF to haze formation, the health effect of high number concentrations of particles with diameters of several or tens of nanometers would be more essential. NPF usually occurs around the same time period as people commute to work. The effects of exposure to a high particle number concentration environment should be investigated.

Atmospheric nucleation and subsequent growth of newly-formed particles could have significant effects on air quality and climate by contributing to CCN (IPCC, 2013). NPF were calculated to enhance CCN number significantly with ratios ranging from 1.2-1.8 in Shanghai in YRD region of China (Leng et al., 2014). Considering both NPF and non-event days, the average contributions of NPF events to potential CCN in the afternoon were calculated to be 11% and 6% at urban sites and regional sites, respectively (Peng et al., 2014). It seems that the enhancement of CCN due to NPF in China at a regional scale was larger than that in Europe (Shen et al., 2016b), which might due to the combination of a higher nucleation rate and quicker subsequent condensable growth associated with higher pollution levels in China. NPF events was also found to have a greater impact on CCN in polluted urban sites than in regional or rural sites in China. For example, CCN number concentrations were observed to be enhanced by a factor of 2-6 in background regions and by a factor of 5.6-8.7 in polluted regions during the NPF event days (Wang et al., 2013c;Shen et al., 2016b). Nevertheless, the impact of NPF on the CCN number concentration was found to depend on the location and individual character of each NPF event, including different hygroscopic properties of particles and thus different CCN activities during different NPF events, so Ma et al. (2016) suggested not using a fixed parameter to predict the contribution of NPF on CCN and Tao et al (2018) emphasized the importance of real-time measurements of hygroscopicity of particles.

During the past 15 years, a lot of NPF observations and related studies were carried in China but, as summarized by Wang et al. (2017), the application of state-of-the-art instruments are still quite limited in China. In recent years, an increasing number of studies utilized more advanced instruments, such as PSM (Xiao et al., 2015;Dai et al., 2017;Yu et al., 2016;Yao et al., 2018), NAIS (Jayaratne et al., 2017;Lv et al., 2018), DEG-SMPS (Cai and Jiang, 2017;Cai et al., 2017) and APi-ToF-MS/CI-APi-ToF-MS (Yao et al., 2018), greatly improving our understanding about the nucleation and particle growth mechanisms in China, especially in highly-polluted environments. However, the lack of continuous and comprehensive long-term observations, which should include measurements of particle number size distribution preferably down to 1–2 nm and vapors that potentially participate in NPF and subsequent particle growth ($H_2SO_4$, ELVOCs, LVOC, ammonia and amines), still limits our understanding on the mechanism of NPF in different environments in China. Key participants and processes of NPF under complex air pollution conditions in China still wait to be answered, and the unexpected NPF at high aerosol loadings need to be explained. Contributions of different mechanisms to NPF should be evaluated with the consideration of spatiotemporal difference, and possibly also with the consideration of inter-annual variability in the process of air pollution control in China. Long periods and comprehensive observations would be the most important factor when investigating NPF mechanisms in China, while laboratory experiments and model simulations would also be very helpful and necessary. As suggested by Kulmala (2018), environmental grand challenges, such as climate change, water and food security as well as urban air pollution, are all linked and need to be studied together. The effects of NPF in China on climate change and human health is still poorly understood and should be evaluated quantitatively. Although a global view is needed for these common challenges of mankind, densely-populated China will no doubt be a very important area in this respect. Studying these effects will be essential for future studies of NPF in China and will be important to a global effort for a better atmosphere on Earth.

## 6. Acknowledgements

This work was supported by Academy of Finland (1251427, 1139656, 296628, 306853, Finnish centre of excellence 1141135), and the EC Seventh Framework Program and European Union's Horizon 2020 program (ERC, project no.742206 "ATM-GTP").

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

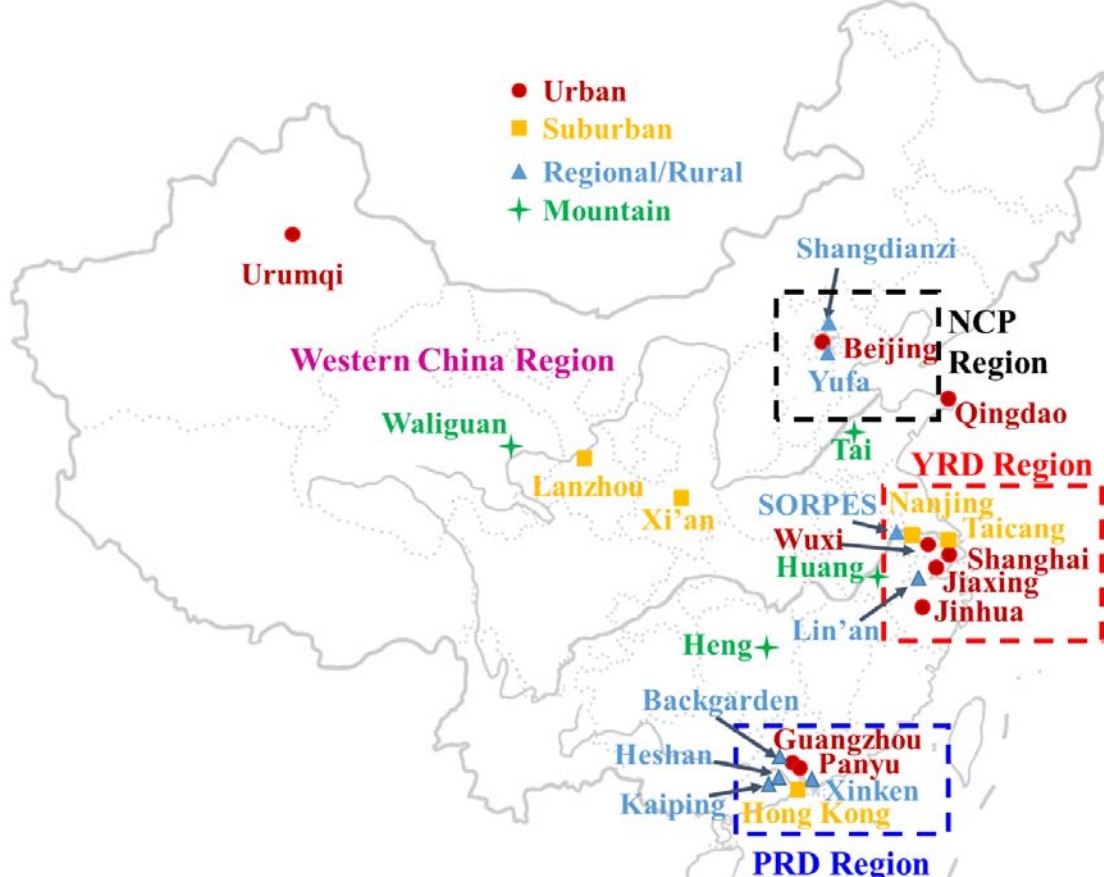

**Figure 1:** Map of observation sites involving NPF study in China. Most of these observations sites were classified into four regions in this study, i.e. North China Plain (NCP), Yangtze River Delta (YRD), Pearl River Delta (PRD), Western China region (Western).

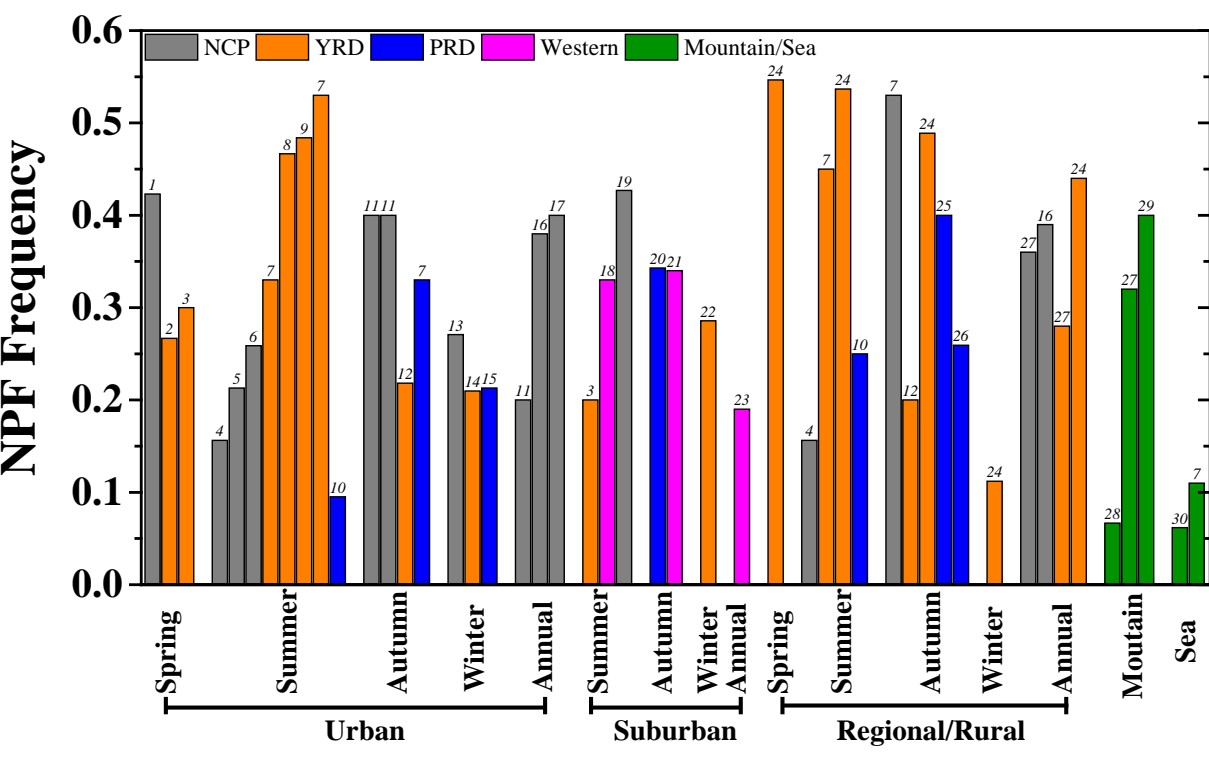

**Figure 2:** NPF frequency observed at different places in different seasons in China. The number on top of each column indicates the references of the data: 1(Cai et al., 2017); 2(Leng et al., 2014); 3(Zhu et al., 2013); 4(Yue

et al., 2009); 5(Zhang et al., 2011); 6(Wang et al., 2011); 7(Peng et al., 2014); 8(Yu et al., 2016); 9(Shen et al., 2016a); 10(Yue et al., 2013), 11(Shen et al., 2016c); 12(Dai et al., 2017); 13(Jayaratne et al., 2017); 14(Xiao et al., 2015); 15(Tan et al., 2016); 16(Wang et al., 2013c); 17(Wu et al., 2007); 18(Gao et al., 2011); 19(Gao et al., 2012); 20(Guo et al., 2012); 21(Zhang et al., 2017b); 22(Herrmann et al., 2014); 23(Peng et al., 2017); 24(Qi et al., 2015); 25(Wang et al., 2013e); 26(Liu et al., 2008); 27(Shen et al., 2016b); 28(Zhang et al., 2017c); 29(Lv et al., 2018); 30(Liu et al., 2014).

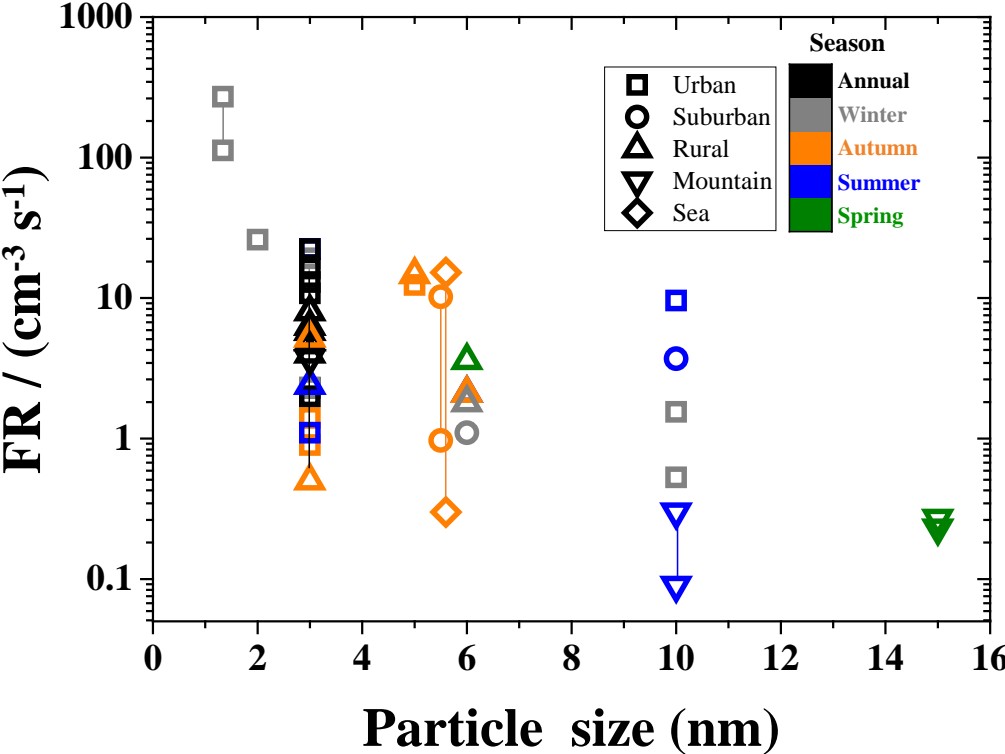

10    **Figure 3:  New particle FR observed at different places in different seasons in China. The line between two data points indicates that a range of FR was reported in the literature. The data are collected in the references in Table 2.**

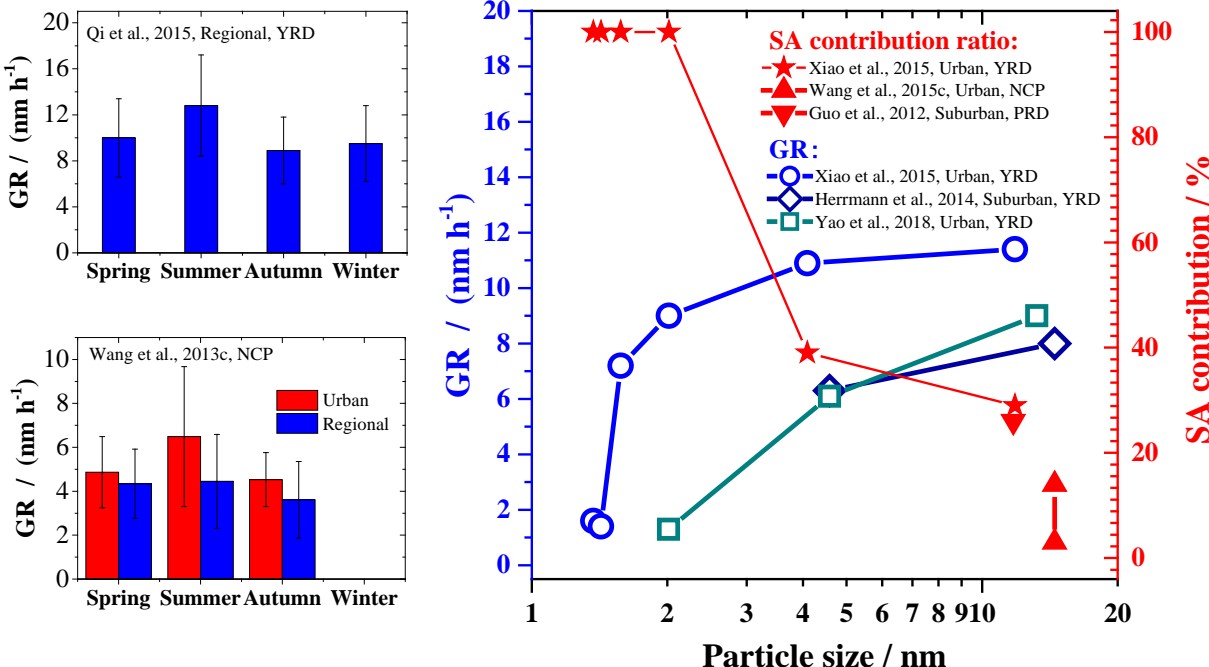

**Figure 4:** Particle GR observed in different seasons at a regional site in YRD (a) and an urban and a regional site in NCP (b) and some measurement results of GR and sulfuric acid (SA) contribution to the GR in different size ranges (c). The data are collected in the references indicated in the figure.

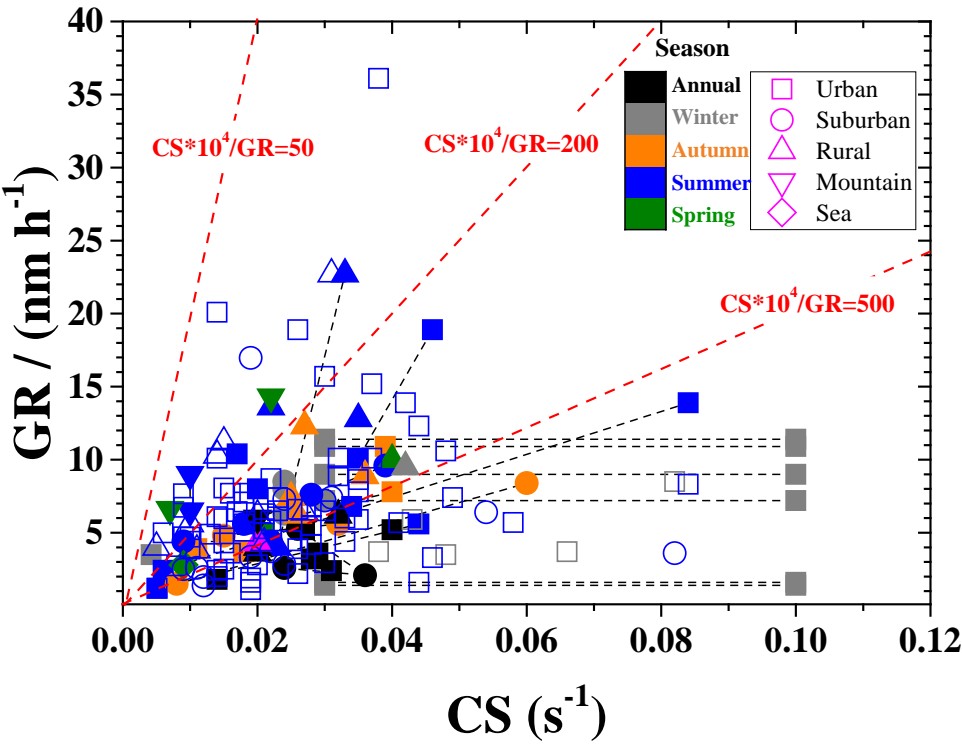

**Figure 5:** Particle GR as a function of CS in the NPF events in China. The solid points are average data for a certain observation period, while the open points are data for individual NPF days. The line between two data points indicates that a range of GR and/or CS was reported in the literature. The data are collected in the references in

10                                                                          **Table 2.**

**Table 1:    Parameters to characterize NPF events**

| Parameter | Description | Calculated from |
|---|---|---|
| **FR** | Formation rate of particles | Temporal variation of particle size distribution |
| **GR** | Growth rate | Temporal variation of particle size distribution |
| **CS** | Condensation sink | Particle size distribution |
| **CoagS** | Coagulation sink | Particle size distribution |
| **$C_{cv}$** | Condensation vapor concentration | GR |
| **Q** | Source rate of condensation vapors | CS and $C_{cv}$ |

**Table 2:** **Characterization of NPF events in China**

| Observation time | Observation site | Region | Site type | Season | EP* | FR (cm⁻³ s⁻¹) | Size for FR (nm) | GR (nm h⁻¹) | Size for GR (nm) | $C_{cv}$ ($10^7$ cm⁻³) | Q ($10^6$ cm⁻³) | CS (s⁻¹) | Measured size (nm) | Ref. | Comments |
|---|---|---|---|---|---|---|---|---|---|---|---|---|---|---|---|
| 2014.7-12 2015.6-8 | Taishan | NCP | Mountain | Summer Autumn | 40% | 7.10 (0.82-25.04) | 3 | 1.98 (0.58-7.76) | 3-20 | | | 0.014 (0.001-0.284) | | (Lv et al., 2018) | |
| 2015.8- 2015.9 | Beijing | NCP | Urban | Annual | | | | 3.2(1.5-6.1)** 3.6(1.4-7.5) | | | | | | (Du et al., 2017) | ground 260m |
| 2015.10- 2016.1 | Beijing | NCP | Urban | Winter | 27% | 26 (12-38) | 2-3 | 3.5(0.5-9) | 2-10 | | | 0.0042 | | (Jayaratne et al., 2017) | |
| 2013.11- 2014.12 | Xi'an | Western | Suburban | Annual | 19% | | | 5±1.9 | 2.8-10.7 | | | | 10-487 | (Peng et al., 2017) | |
| 2018.4-7 | Huang | YRD | Mountain | Summer | 7% | 0.09-0.3 | 10 | 2.90(1.42-4.53) | | | | | 10-10000 | (Zhang et al., 2017c) | |
| 2014.8-11 | Lanzhou | Western | Suburban | Autumn | 34% | 1.71 | | 6.1 | | | | | 14.6-661.2 | (Zhang et al., 2017b) | |
| 2016.3-4 | Beijing | NCP | Urban | Spring | 42% | 22-156*** | 1.5 | 1.2-3.3 | | | | | 1- | (Cai et al., 2017) | $H_2SO_4$ measured |
| 2015.9-11 | Nanjing | YRD | Urban | Autumn | 22% | >1000*** | 1.3 | >20 | <3 | | | 0.04 | | (Dai et al., 2017) | |
| | | | | | | 2.5-27.2 | 5 | 7.8 | 10 | | | 0.04 | | | |
| | YRD | YRD | Regional | Autumn | 20% | 8.0-24.1 | 5 | 6.2 | 10 | | | 0.026 | | | |
| 2014 Youth Olympic | Nanjing | YRD | Urban | Summer | 47% | 92-2500*** | 1.42 | 1.6-8.9 | 1.4-3 | | | | 1.4- | (Yu et al., 2016) | |
| 2011.11-12 | Panyu, Guangzhou | PRD | Urban | Winter | 21% | 0.97 | 10-20 | 8.5 | | | 9.7 | 0.082 | 10-20000 | (Tan et al., 2016) | |
| | | | | | | 0.82 | | 5.9 | | | 3.5 | 0.043 | | | |
| | | | | | | 1.55 | | 3.7 | | | 1.9 | 0.038 | | | |
| | | | | | | 0.53 | | 3.5 | | | 2.3 | 0.048 | | | |
| | | | | | | 0.57 | | 3.7 | | | 3.4 | 0.066 | | | |
| 2008.8-9 | Beijing | NCP | Urban | Autumn | 40% | 0.9 | 3-10 | 3.7 | 3-10 | | | 0.018 | | (Shen et al., | |

| Observation time | Observation site | Region | Site type | Season | EP* | FR (cm$^{-3}$ s$^{-1}$) | Size for FR (nm) | GR (nm h$^{-1}$) | Size for GR (nm) | $C_{cv}$ (10$^7$ cm$^{-3}$) | Q (10$^6$ cm$^{-3}$) | CS (s$^{-1}$) | Measured size (nm) | Ref. | Comments |
|---|---|---|---|---|---|---|---|---|---|---|---|---|---|---|---|
| 2010-2013 | | | | Annual | 20% | 2.3 | | 5.2 | | | | 0.04 | | 2016c | |
| 2015.8-9 | | | | Autumn | 40% | 1.4 | | 3.9 | | | | 0.011 | | | |
| 2008.3–2013.12 | Shangdianzi | NCP | Rural | Annual | 36% | 6.3(0.5-39.3) | | 3.6(0.7-13.4) | | | | 0.02 | | | |
| 2011.1–2011.12 | Taishan | NCP | Mountain | Annual | 32% | 3.7(1-9.6) | 3 | 6(1.1-15.4) | | | | 0.02 | 3-800 | (Shen et al., 2016b) | |
| 2013.1–2013.12 | Lin an | YRD | Rural | Annual | 28% | 5.8(0.8-26.5) | | 6.2(1.8-21.3) | | | | 0.032 | | | |
| 2015.5 | Jiaxing | YRD | Urban | Summer | 48% | 9.6(4-17) | 10-20 | 6.8(2.2-15.7) | | 9.4(3.0-21.5) | 3.3(0.5-7.7) | 0.034(0.015-0.058) | 10-10000 | (Shen et al., 2016a) | |
| 2008.7-9 | Beijing | NCP | Urban | Summer | | | | | | | | | 3-900 | (Wang et al., 2015c) | H$_2$SO$_4$, VOCs measured |
| 2013.11-2014.1 | Shanghai | YRD | Urban | Winter | 21% | 112.4-271 | 1.34 | 1.6±1 | 1.35-1.39 | 2.3-6.4 | | 0.03-0.1 | 1.35-615 | (Xiao et al., 2015) | |
| | | | | | | 2.3-19.2 | 3 | 1.4±2.2 | 1.39-1.46 | | | | | | |
| | | | | | | | | 7.2±7.1 | 1.46-1.7 | | | | | | |
| | | | | | | | | 9±11.4 | 1.7-2.39 | | | | | | |
| | | | | | | | | 10.9±9.8 | 2.39-7 | | | | | | |
| | | | | | | | | 11.4±9.7 | 7-20 | | | | | | |
| 2011-2013 | YRD | YRD | Regional | Annual | 44% | | | | | | | 0.038-0.053 | 6-800 | (Qi et al., 2015) | |
| | | | | Spring | 55% | 3.6 | | 10 | | | | 0.04-0.055 | | | |
| | | | | Summer | 54% | 2.1 | 6 | 12.8 | 6-30 | | | 0.035-0.05 | | | |
| | | | | Autumn | 49% | 2.1 | | 8.9 | | | | 0.036-0.051 | | | |
| | | | | Winter | 11% | 1.8 | | 9.5 | | | | 0.042-0.058 | | | |
| 2011 | Yellow River Delta | NCP | | Annual | 22% | 6.6 | | 5.3 | | | | | | (Yuan et al., 2015) | |

| Observation time | Observation site | Region | Site type | Season | EP* | FR $(cm^{-3} s^{-1})$ | Size for FR (nm) | GR $(nm\ h^{-1})$ | Size for GR (nm) | $C_{cv}$ $(10^7 cm^{-3})$ | Q $(10^6 cm^{-3})$ | CS $(s^{-1})$ | Measured size (nm) | Ref. | Comments |
|---|---|---|---|---|---|---|---|---|---|---|---|---|---|---|---|
| 2012.7-8 | Nanjing | YRD | Suburban | Summer | | 3.7(1.6-6.7) | 10 | 7.6(5.6-9.6) | 10-20 | 10.5(7.7-13.2) | 2.9(1.9-4.7) | 0.028(0.018-0.039) | | (An et al., 2015) | |
| 2010.7-8 | Wuxi | YRD | Urban | Summer | 53% | | | 10.4(6.2-13.3) | | | | 0.017(0.009-0.029) | | | |
| 2010.4-6 | Shanghai | YRD | Urban | Summer | 33% | | | 8(4.2-12) | | | | 0.02(0.01-0.033) | | | |
| 2010.11 | Guangzhou | PRD | Urban | Autumn | 33% | | | 10.9(7.3-18.1) | | | | 0.039(0.026-0.056) | | | |
| 2008.5-6 | Urumchi | Western | Urban | Spring | 86% | | | | | | | 0.016(0.01-0.026) | | | |
| 2010.6 -7 | Jiaxing | YRD | Regional | Summer | 45% | | | 13.6(7.9-19.6) | | | | 0.022(0.011-0.041) | | (Peng et al., 2014) | |
| 2007.10 | Yufa | NCP | Regional | Autumn | 53% | | | 12.3(8.6-21) | | | | 0.027(0.005-0.053) | | | |
| 2008.10-11 | Kaiping | PRD | Regional | Autumn | 40% | | | 7.4(3.2-13.5) | | | | 0.025(0.003-0.086) | | | |
| 2009.10-12 | BG | PRD | Sea | Autumn | 15% | | | 4.5(3.2-7.5) | | | | 0.014(0.01-0.018) | | | |
| 2011.3-4 | Changdao | NCP | Sea | Spring | 19% | | | 5.7(4.5-6.8) | | | | 0.02(0.019-0.021) | | | |
| 2011.10-11 | Wenling | YRD | Sea | Autumn | 10% | | | 7.5 | | | | 0.026 | | | |
| 2011.3-4 | China sea | | Sea | Spring | 11% | | | 2.8(1.6-3.9) | | | | 0.009(0.008-0.011) | | | |
| 2011.10-11, 2012.11 | Marginal seas | | Sea | Autumn | 6% | 0.3-15.2 | 5.6-30 | | | | 1.6(1.1-2.2) | | | (Liu et al., 2014) | |
| 2010.5 | Qingdao | NCP | Coastal city | Summer | 41% | 13.3 | | 6.2 | | | | | | (Zhu et al., 2014) | |
| 2011.8-9 | Huangshan | YRD | Mountain | Summer | | 0.25-0.67 | 10-20 | 6.5-9.0 | 10-20 | | | 0.01 | | (Wang et al., 2014b) | |
| 2011.10-11 | Nanjing | YRD | Suburban | Autumn | | 0.83-1.67 | | 4.8-5.6 | | | | 0.015-0.032 | | | |
| 2011.11-2012.3 | Nanjing | YRD | Suburban | Winter | 29% | 1.1 | 6 | 8.50 / 6.3 / 8 | 6-30 / 3-7 / 7-30 | | 3.8 | 0.024 | 0.8-800 | (Herrmann et al., 2014) | |
| 2012.4 | Shanghai | YRD | Urban | Spring | 27% | 0.40 | | 4.91 | | | | 0.021 | 10-800 | (Leng et al., 2014) | |
| 2009.4 | Mountain in Hunan | | Mountain | Spring | | 0.27 | 15 | 14.3 | 30-50 | | 2.8 | 0.007 | 10-10000 | (Nie et al., 2014) | dust |
| 2009.4 | Mountain in Hunan | | Mountain | Spring | | 0.23 | 15 | 6.6 | 30-50 | | 1.4 | 0.022 | 10-10000 | (Nie et al., 2014) | non-dust |

| Observation time | Observation site | Region | Site type | Season | EP* | FR (cm$^{-3}$ s$^{-1}$) | Size for FR (nm) | GR (nm h$^{-1}$) | Size for GR (nm) | $C_{cv}$ (10$^7$ cm$^{-3}$) | Q (10$^6$ cm$^{-3}$) | CS (s$^{-1}$) | Measured size (nm) | Ref. | Comments |
|---|---|---|---|---|---|---|---|---|---|---|---|---|---|---|---|
| 2006.7 | BG | PRD | Rural | Summer | 25% | 2.4-4 | 3-25 | 4-22.7 | | 5-31 | 1.3-11 | 0.023-0.033 | 15-10000 | (Yue et al., 2013) | |
| | Guangzhou | PRD | Urban | Summer | 10% | NA | | 10.1-18.9 | | 14-26 | 7.7-9.1 | 0.035-0.046 | 3-10000 | | |
| 2008.10-11 | Kaiping | PRD | Rural | Autumn | 40% | | | 7.4 / 3.2-135.5 | | 1.3 (H$_2$SO$_4$) | | 0.025 / 0.03-0.086 | | (Wang et al., 2013e) | |
| 2008.3-11 | Beijing | NCP | Urban | Annual | 38% | 10.8(2.2-34.5) | | 5.2 | | 9.3 | 2.1 | 0.027 | | (Wang et al., 2013c) | |
| | Shangdianzi | NCP | Regional | Annual | 39% | 4.9(0.4-24.5) | | 4 | | 7.1 | 1.2 | 0.02 | | | |
| 2010.10-11 | HK | PRD | Suburban | Autumn | 34% | 0.97-10.2 | 5.5 | 1.5-8.4 | | 0.8-1.2 (H$_2$SO$_4$) | | 0.008-0.06 | 5.5-350 | (Guo et al., 2012) | |
| 2010 summer | Nanjing | YRD | Suburban | Summer | 20% | | | | | | | | | (Zhu et al., 2013) | |
| 2009 autumn | | | Urban/Suburban | Autumn | 3% | | | 10-16 | | | | | | | |
| 2011 spring | | | Urban | Spring | 30% | | | 6.8-8.3 | | | | | | | |
| 2008 | Beijing | NCP | Suburban | Summer | 43% | | | 3.2(1.2-8) | | 4.4 | 1.2 | | 10-1000 | (Gao et al., 2012) | |
| 2006.6-7 | Lanzhou | Western | Suburban | Summer | 33% | 1.8-3.4 | 10-20 | 4.4(1.3-16.9) | | 6.1 | 1.1 | 0.009-0.022 | 10-10000 | (Gao et al., 2011) | |
| 2008.10-2009.2 | Shanghai | YRD | Urban | Winter | 5% | 0.2-0.5 | 10-20 | 3.3-5.5 | | | | | 10-10000 | (Du et al., 2012) | |
| 2004-2006 | Beijing | NCP | Urban | Annual | | | | 0.4 | | 0.6 (H$_2$SO$_4$) | | 0.01 | 3-10000 | (Wu et al., 2011) | |
| 2008.7-9 | Beijing | NCP | Urban | Summer | 26% | | | | | | | 0.022 | 3-900 | (Wang et al., 2011) | H$_2$SO$_4$ measured |

| Observation time | Observation site | Region | Site type | Season | EP* | FR $(cm^{-3}\,s^{-1})$ | Size for FR (nm) | GR $(nm\,h^{-1})$ | Size for GR (nm) | $C_{cv}$ $(10^7\,cm^{-3})$ | Q $(10^6\,cm^{-3})$ | CS $(s^{-1})$ | Measured size (nm) | Ref. | Comments |
|---|---|---|---|---|---|---|---|---|---|---|---|---|---|---|---|
| 2008.3-2009.8 | Shangdianzi | NCP | Rural | Annual | 36% | 8(0.7-72.7) | | 4.3(0.3-14.5) | | | | 0.02 | 3-10000 | (Shen et al., 2011) | |
| 2008.6-9 | Beijing | NCP | Urban | Summer | 21% | | | 2.43-13.9 | | | | 0.006-0.084 | 12-550 | (Zhang et al., 2011) | |
| 2008 | Beijing | NCP | Urban | Summer | 10-70% | 2-13 | | 3-11 | | | | | 3-900 | (Yue et al., 2010) | |
| 2003-2004 | HK | PRD | Coastal/Suburban | Annual | | 3.6(2.3-11.5) | | 2-11.8 | | | | | 3.2-106 | (Yao et al., 2010) | |
| 2006 | Beijing | NCP | Urban | Summer | 16% | 1.1-22.4 | | 1.2-5.6 | | 1.5-17 | 0.1-8.8 | 0.0051-0.044 | 3-10000 | (Yue et al., 2009) | |
| | Yufa | NCP | Rural | Summer | 16% | | | | | | | | | | |
| 2005.5-6 | Taicang, YRD | YRD | Suburban | Summer | | | | 6.4(3.6-7.4) | | | | | 10-1000 | (Gao et al., 2009) | |
| 2004 | Xinken | PRD | Rural | Autumn | 26% | 0.5-5.2 | | 2.2-19.8 | | | | | 3-10000 | (Liu et al., 2008) | |
| 2005-2006 | China Sea | | Sea | Annual | | | | 3.4 | | | | | 15-10000 | (Lin et al., 2007) | |
| | | | | | | | | 3.5 | | | | | | | |
| 2004-2005 | Beijing | NCP | Urban | Annual | 40% | 3.3-81.4 | | 0.1-11.2 | | | | | 3-10000 | (Wu et al., 2007) | |
| | | | | | | 22.3 | | 1.8 | | 2.5 | 0.59 | 0.014 | | | clean |
| | | | | | | 16.2 | | 2.4 | | 6 | 2.4 | 0.031 | | | pollution |

*EP: events percentage (frequency).

**The average value and the range of the reported data in the parenthesis.

***These formation rates are the maximum FR during the NPF event.