# Peer review of "Atmospheric new particle formation in China"

_Atmospheric Chemistry and Physics, 2018_

## Referee Comment (RC1) · Anonymous Referee #1 · 25 Sep 2018

This manuscript summarizes the atmospheric new particle formation (NPF) studies in China currently available in the literature, which represents a major effort in advancing our understanding on NPF in China. A number of NPF parameters, including frequency, formation rate, and growth rate, have been compared across the country. In addition, NPF under a heavily polluted atmosphere is discussed. From this perspective, this manuscript should be published in Atmos. Chem. Phys. On the other hand, this reviewer believes that the authors should be more critical and try to rationalize NPF features in China according to the latest research findings. Below are the detailed comments.

1. The authors provide an excellent summarization of NPF papers in the literature, but it is not a critical review. The senior authors are the leading figures in current NPF research, and I believe, they are perfectly aware of the drawbacks of some of the research that have been conducted, as they have acknowledged in the manuscript. For

[Figure]

example, NPF measurements with a 10 nm detection limit may overestimate the NPF frequency, because the growth of the automobile emission of sub-10 nm particles may look like a NPF event. I fully understand that it is premature to judge an ambient observation, but there are potentially two strategies for this issue. The authors can only include journal publications that are mostly related to the topic, but looks to me that this is not what the authors want to do according to the current format of the manuscript. On the other hand, the authors can at least state the latest findings/conclusions, caution the readers, and ask the readers to be selective and look into the references. Also, my impression after reading the current manuscript is that there are many, many possibilities in the observed events. Can the authors help to rule out some, which is the value of a critical review?

2. Key publications are missing. Included are, but not limited to, measurements of amines by Lin WANG's group at Fudan University, the ammonia network by Yuesi Wang's group at Institute of Atmospheric Physics, and HOMs' role in NPF by Aijun Ding's group at Nanjing University.

3. Figure 2 could be misleading. Some of the formation rates are the daily averages whereas some are maximum values during a day.

4. (Figure 3), there are so many data points are color-coded with gray, which, as stated by the authors, come without a size information. I would rather remove these gray points because they make the figure pretty busy and no clue. Well, the figure is still quite obscure even without the gray data points. On the other hand, what does "GR size" really stand for? The upper size, the lower size, or something else?

5. (Figure 4), GR could vary by a factor of 10 for different particle size ranges. I suspect that the current GR for the data points in Figure 4 are in fact values in a wide range of particle sizes. This would not make sense if one plots in this way.

Minor comments, 6. (Page 1, Line 9), is the goal of "exploring the nucleation and particle growth mechanisms" achieved? 7. (Page 1, Line 11), revise "cannot not be

fully explain" 8. (Page 1, Line 17), rephrase the sentence 9. (Page 1, Line 20), in terms of number concentrations 10. (Page 1, Line 24), rephrase "gaseous vapors nucleation" 11. (Page 1, Line 29), HOMs instead of HOM 12. (Page 2, Line 25), revise "recentZu years" 13. (Page 2, Line 28), rephrase "which was later than that observed for SO2" 14. (Page 3, Line 12), revise "including in" 15. (Page 3, Line 17), The authors stated here that they summarized NPF studies in polluted regions in China. In fact, many included studies are from clean atmospheres, even compared to European countries 16. (Page 3, Line 26), provide evidence that Xi'an and Urumqi are more polluted. Is this only from one study or always true? In fact, a yearly average might be more convincing. 17. (Page 3, Line 29-), provide the detailed locations of the sites mentioned, since readers may not be familiar with these Chinese supersites. Also the same for Table 2. 18. (Session 3.1), In addition to the many factors that have been discussed in the manuscript, emission is a major player in determining the NPF frequency, which is not quite emphasized in the current manuscript. 19. (Session 3.2), discuss Kerminen-Kulmala equation under the umbrella of Kulmala et al. Faraday Discussions 2018. Deriving J1 from particle formation rate at a larger size may not be feasible. 20. (Page 8, Line 19), rephrase "in the NPF Chinese megacities" 21. (Page 9, Line 15-19), rephrase the sentence 22. (Page 9, Line 32), Tao et al. (2016) looked at really large particles, the composition of which may not be directly related to NPF 23. (Figure 2), assign the references to each of the bars, either in the figure or a number in the figure and details in the figure caption

---

## Referee Comment (RC2) · Anonymous Referee #2 · 30 Oct 2018

The authors summarized the data of NPF measurements reported in China in the past 14 years. The features of NPF parameters such as frequency, FR and GR were described, and the possible reasons behind them were also discussed. The author raised attention on the ambiguous relationship between NPF and haze formation, and pointed out the importance of comprehensive study including laboratory works and field measurement on NPF research in China. The data set of this paper is full accurate, could be helpful for other researchers. However, there are still some concerns to be satisfied before I suggest the paper could be published on ACP.

Major suggestions: In Section 3, the authors made so far most complete and detailed summary of NPF field measurement results in China. However, on scope of scientific issues, similar understandings were already raised by the review papers from Wang et al, Kulmala et al and Kerminen et al. The authors should strengthen the highlight of this paper by:

[Figure]

1. Emphasizing the new findings from studies published in recent years, utilizing the up-to-date techniques, e.g. measurements of sub-3 nm particles and sulfuric acid, since these were not included in former papers.

2. Enriching the discussion in relationship between haze and NPF in Section 4, concentrating on issues such as : 1) threshold of the occurrence of NPF; 2) contribution of NPF on haze formation, in comparison with primary emissions.

Minor suggestions:

1. Page2 Line 25 "recentZu years" Type error.

2. Page3 Line 26 The air pollution level decreases from NCP to PRD, or from north to south, among these three regions. References are needed.

3. Page5 About regional occurrence of NPF, Shen et al's work is encouraged to be cited.

Shen, X. J., Sun, J. Y., Kivekas, N., Kristensson, A., Zhang, X. Y., Zhang, Y. M., Zhang, L., Fan, R. X., Qi, X. F., Ma, Q. L., and Zhou, H. G.: Spatial distribution and occurrence probability of regional new particle formation events in eastern China, Atmospheric Chemistry And Physics, 18, 587-599, 10.5194/acp-18-587-2018, 2018.

4. Page6 Season and the origin of air mass make impact on occurrence of NPF by influencing the level of CS and precursors. The suggestion is moving the discussions on them after those on CS and gaseous pollutants.

5. Page7 CS/GR may not be a good index on the probability of NPF, because GR couldn't represent the level of nucleation precursor. The compounds that are crucial for nucleation (e.g. $H_2SO_4$, amine) may be negligible in particle growth. The suggestion is providing a review the value of $SO_2*UVB/CS$ in references, since the measurement of $H_2SO_4$ was quite limited in China.

Kerminen, V. M., Chen, X. M., Vakkari, V., Petaja, T., Kulmala, M., and Bianchi, F.:

Atmospheric new particle formation and growth: review of field observations, Environ. Res. Lett., 13, 38, 10.1088/1748-9326/aadf3c, 2018.

Kulmala, M., Kerminen, V. M., Petaja, T., Ding, A. J., and Wang, L.: Atmospheric gas-to-particle conversion: why NPF events are observed in megacities?, Faraday Discussions, 200, 271-288, 10.1039/c6fd00257a, 2017.

Wang, Z., Wu, Z., Yue, D., Shang, D., Guo, S., Sun, J., Ding, A., Wang, L., Jiang, J., Guo, H., Gao, J., Cheung, H. C., Morawska, L., Keywood, M., and Hu, M.: New particle formation in China: Current knowledge and further directions, The Science of the total environment, 577, 258-266, 10.1016/j.scitotenv.2016.10.177, 2017.
* * ** * *

---

## Author Comment (AC1) · 2 Dec 2018

**Author Final Response**
**Ms. Ref. No.: acp-2018-612**
**Title: "Atmospheric new particle formation in China"**

We appreciate the comments from the reviewers on this manuscript. We have answered them in the following paragraphs (the text in italics is the reviewer comments, followed by our response) point by point. The line numbers in the response are from the revised manuscript.

**Response for Reviewer #1**

*This manuscript summarizes the atmospheric new particle formation (NPF) studies in China currently available in the literature, which represents a major effort in advancing our understanding on NPF in China. A number of NPF parameters, including frequency, formation rate, and growth rate, have been compared across the country. In addition, NPF under a heavily polluted atmosphere is discussed. From this perspective, this manuscript should be published in Atmos. Chem. Phys. On the other hand, this reviewer believes that the authors should be more critical and try to rationalize NPF features in China according to the latest research findings. Below are the detailed comments.*

*1. The authors provide an excellent summarization of NPF papers in the literature, but it is not a critical review. The senior authors are the leading figures in current NPF research, and I believe, they are perfectly aware of the drawbacks of some of the research that have been conducted, as they have acknowledged in the manuscript. For example, NPF measurements with a 10 nm detection limit may overestimate the NPF frequency, because the growth of the automobile emission of sub-10 nm particles may look like a NPF event. I fully understand that it is premature to judge an ambient observation, but there are potentially two strategies for this issue. The authors can only include journal publications that are mostly related to the topic, but looks to me that this is not what the authors want to do according to the current format of the manuscript. On the other hand, the authors can at least state the latest findings/conclusions, caution the readers, and ask the readers to be selective and look into the references. Also, my impression after reading the current manuscript is that there are many, many possibilities in the observed events. Can the authors help to rule out some, which is the value of a critical review?*

**Response:** Thanks for the reviewer's comments. We went through the whole manuscript and reorganized the summary of results of some references. Some critical comments and latest findings/conclusions were added. Some examples of these revisions are listed here:

Page 4, lines 17-19: "The relative short-period observations may not represent varying atmospheric conditions, and therefore, the applicability of these observation results may be limited to specific conditions."

Page 5 line 26- page 6 line 12: "A challenge that exists for identifying NPF is the interference of primarily emitted particles from local combustion sources near the observation site. For example, the formation and rapid growth of vehicular particles during the initial 1–2 s of exhaust cooling and dilution processes frequently lead to a nucleation mode at 10–20 nm (Vu et al., 2015;Lee et al., 2015). Spikes of particle number concentration associated with combustion emissions were observed in many NPF studies (Liu et al., 2014;Wang et al., 2014;Peng et al., 2017;Zhu et al., 2017), but these spikes usually had some different characteristics from those of the NPF events (Wang et al., 2014). The particle size (Hofman et al., 2016), the ratio of particle number concentrations in the nucleation mode particles to those of fine particles (Peng et al., 2017;Jung et al., 2013), the time of duration of NPF events (Zhu et al., 2017), and the correlation of the particle number concentration with other gaseous pollutants concentrations and meteorology conditions (Wang et al., 2014) were used to identify the contribution of primary emission in the burst of particle number concentration. However, there are still uncertainties in distinguishing the new-particle signal from the mixed signals of newly-formed particles and freshly-emitted particles from combustion, especially when NPF measurements were carried out with a particle size detection limit larger than 10 nm. There is a possibility that the growth of the vehicular emission of sub-10 nm particles may look like a NPF event and therefore overestimate the NPF frequency. A recently observation found a notable presence of traffic-originated nanocluster aerosol particles in the size range of 1.3–3.0 nm in urban air (Rönkkö et al., 2017), which might raise new questions about the sources of nanocluster aerosol particles in semi-urban roadside environments."

Page 6 lines 21-22: "Shen et al. (2018) observed regional NPF in NCP with a horizontal extent larger than 500 km and found that large-scale regional NPF was favored by a fast transport of northwesterly air masses."

Page 6 lines 29-31: "Despite different pollution conditions in different regions of China, there is a lack of long-term NPF observations, which limits our knowledge about the relationship between the level of air pollution and the occurrence of NPF."

Page 7 lines 19-20: "However, higher $SO_2$ concentrations could increase the probability of occurrence of NPF events at a mountaintop site (Lv et al., 2018)."

Page 8 lines 2-6: "In spite of an increasing number of aerosol size distribution measurements in China, atmospheric NPF observations that cover the full annual cycle are still quite limited. Meanwhile, the main reason for the different NPF event frequencies in different seasons is still uncertain because many factors influencing NPF, such as the radiation intensity, temperature, relative humidity, wind properties, biogenic activity and anthropogenic emissions, tend to be changed simultaneously."

Page 8 lines 20-24: "These results highlighted the complex relationship between air pollution and NPF. Many factors, including pre-existing aerosols, organic pollutants and $SO_2$, are connected each other due to their similar emission sources, so it is not easy to extract the influence of one factor on NPF. Furthermore, since environments are complex and diverse, some other factors, such as the concentration of OH radicals and topography, can also be important to NPF and therefor deserve further investigation in both field observations and controlled experiments."

Page 8 lines 29-page 9 line 1: "Besides, the assumed coagulation sticking probability of 1 for molecular clusters with pre-existing particles in their collision and the unclear GR of sub-3 nm particles might result in errors in the derivation of FR (Kulmala et al., 2017)"

Page 9 lines 18-19: "although nucleation rates were rarely calculated using measurements of particles in the size range of 1-3 nm."

Page 11 lines 16-19: "Yu et al. (2016) estimated that a high concentration of extremely low volatility organic compounds was the key factor leading to a maximum in GR for very small particles (1.4-3 nm) in urban Nanjing. Although the existence of local maxima in GR in the sub-3 nm size range is highly sensitive to uncertainties in particle size distributions, the results highlighted that detailed investigations for the mechanisms of the initial growth steps of atmospheric NPF are needed (Yu et al., 2016)."

*2. Key publications are missing. Included are, but not limited to, measurements of amines by Lin WANG's group at Fudan University, the ammonia network by Yuesi Wang's group at Institute of Atmospheric Physics, and HOMs' role in NPF by Aijun Ding's group at Nanjing University.*

**Response:** Thanks for the reminding. We have added these studies in the revised manuscript. Besides the publications listed by the reviewer, we also added some other measurements of ammonia by Zhaoyang Meng's group, measurements of amines by Jun Zheng's group, and simulation of amines by Fangqun Yu's group. These work were added in Page 3 lines 4-5, Page 9, lines 27-30, and so on.

Some additional description of the observation results was also added together with the added publications:

Page 9 lines 27- Page 10 lines 4: "According to the national ammonia observation network, the overall average concentration of ammonia in China is much higher than the values observed in the U.S. The seasonal maximum $NH_3$ concentrations were observed in the summer and the most abundant concentrations of $NH_3$ were observed in the NCP region in China (Pan et al., 2018). Compare to $NH_3$, the amine measurements are more sparse (Zheng et al., 2015;Yao et al., 2016), and direct information on amine emissions is currently not available but these emissions have to be estimated by assuming a fixed ratio or source-dependent ratios of amines to total ammonia emissions in China (Mao et al., 2018). Dai et al.(2017) proposed that plumes containing high concentrations of ammonia, amines or HOM produced from their observed VOCs led to strong local NPF events. The observations made at the SORPES station in YRD indicated that HOM played an essential role in the initial condensational growth of newly formed clusters (Huang et al., 2016;Ding et al., 2016;Qi et al., 2018)."

List of the added publications:

Meng, Z. Y., Lin, W. L., Jiang, X. M., Yan, P., Wang, Y., Zhang, Y. M., Jia, X. F., and Yu, X. L.: Characteristics of atmospheric ammonia over Beijing, China, Atmos. Chem. Phys., 11, 6139-6151, 10.5194/acp-11-6139-2011, 2011.

Pan, Y. P., Wang, Y. S., Tang, G. Q., and Wu, D.: Wet and dry deposition of atmospheric nitrogen at ten sites in Northern China, Atmos. Chem. Phys., 12, 6515-6535, 10.5194/acp-12-6515-2012, 2012.

Yao, L., Wang, M. Y., Wang, X. K., Liu, Y. J., Chen, H. F., Zheng, J., Nie, W., Ding, A. J., Geng, F. H., Wang, D. F., Chen, J. M., Worsnop, D. R., and Wang, L.: Detection of atmospheric gaseous amines and amides by a high-resolution time-of-flight chemical ionization mass spectrometer with protonated ethanol reagent ions, Atmos. Chem. Phys., 16, 14527-14543, 10.5194/acp-16-14527-2016, 2016.

Huang, X., Zhou, L., Ding, A., Qi, X., Nie, W., Wang, M., Chi, X., Petäjä, T., Kerminen, V. M., Roldin, P., Rusanen, A., Kulmala, M., and Boy, M.: Comprehensive modelling study on observed new particle formation at the SORPES station in Nanjing, China, Atmos. Chem. Phys., 16, 2477-2492, 10.5194/acp-16-2477-2016, 2016.

Ding, A. J., Nie, W., Huang, X., Chi, X. G., Sun, J. N., Kerminen, V. M., Xu, Z., Guo, W. D., Petaja, T., Yang, X. Q., Kulmala, M., and Fu, C. B.: Long-term observation of air pollution-weather/climate interactions at the SORPES station: a review and outlook, Frontiers of Environmental Science & Engineering, 10, 15, 10.1007/s11783-016-0877-3, 2016.

Meng, Z. Y., Xu, X. B., Lin, W. L., Ge, B. Z., Xie, Y. L., Song, B., Jia, S. H., Zhang, R., Peng, W., Wang, Y., Cheng, H. B., Yang, W., and Zhao, H. R.: Role of ambient ammonia in particulate ammonium formation at a rural site in the North China Plain, Atmos. Chem. Phys., 18, 167-184, 10.5194/acp-18-167-2018, 2018.

Pan, Y. P., Tian, S. L., Zhao, Y. H., Zhang, L., Zhu, X. Y., Gao, J., Huang, W., Zhou, Y. B., Song, Y., Zhang, Q., and Wang, Y. S.: Identifying Ammonia Hotspots in China Using a National Observation Network, Environ. Sci. & Technol., 52, 3926-3934, 10.1021/acs.est.7b05235, 2018.

Mao, J. B., Yu, F. Q., Zhang, Y., An, J. Y., Wang, L., Zheng, J., Yao, L., Luo, G., Ma, W. C., Yu, Q., Huang, C., Li, L., and Chen, L. M.: High-resolution modeling of gaseous methylamines over a polluted region in China: source-dependent emissions and implications of spatial variations, Atmos. Chem. Phys., 18, 7933-7950, 10.5194/acp-18-7933-2018, 2018.

Qi, X. M., Ding, A. J., Roldin, P., Xu, Z. N., Zhou, P. T., Sarnela, N., Nie, W., Huang, X., Rusanen, A., Ehn, M., Rissanen, M. P., Petaja, T., Kulmala, M., and Boy, M.: Modelling studies of HOMs and their contributions to new particle formation and growth: comparison of boreal forest in Finland and a polluted environment in China, Atmos. Chem. Phys., 18, 11779-11791, 10.5194/acp-18-11779-2018, 2018.

*3. Figure 2 could be misleading. Some of the formation rates are the daily averages whereas some are maximum values during a day.*

**Response:** Thanks for pointing out this. The peak (maximum) formation rates of particles in the NPF were removed or replaced by the averages in Figure 3 in the revised manuscript.

*4. (Figure 3), there are so many data points are color-coded with gray, which, as stated by the authors, come without a size information. I would rather remove these gray points because they make the figure pretty busy and no clue. Well, the figure is still quite obscure even without the gray data points. On the other hand, what does "GR size" really stand for? The upper size, the lower size, or something else?*

**Response:** Figure 4 was revised. In the revised version, we gave up to draw a whole picture for all the reported GR values because that would be quite obscure, as the reviewer pointed out. We selected several references and tried to indicate the differences in GR during different seasons and for particles with different sizes.

[Figure]

**Figure 4: Particle GR observed in different seasons at a regional site in YRD (a) and an urban and a regional site in NCP (b) and some measurement results of GR and sulfuric acid (SA) contribution to the GR in different size ranges (c). The data are collected in the references indicated in the figure.**

*5. (Figure 4), GR could vary by a factor of 10 for different particle size ranges. I suspect that the current GR for the data points in Figure 4 are in fact values in a wide range of particle sizes. This would not make sense if one plots in this way.*

**Response:** As mentioned above, Figure 4 was revised. The particle size range for calculating GR was considered.

*Minor comments,*

*6. (Page 1, lines 9), is the goal of "exploring the nucleation and particle growth mechanisms" achieved?*

**Response:** We respond to this with some summary of the current understanding for the nucleation and particle growth mechanisms on page 16 lines 1-9 in the revised manuscript. We concluded that a great progress was made for the nucleation and particle growth mechanisms in China in the past decade, but the answers to some key questions are still lacking.

*7. (Page 1, lines 11), revise "cannot not be fully explain"*
**Response:** It was revised to "cannot fully explain".

*8. (Page 1, lines 17), rephrase the sentence*
**Response:** It was revised to "In addition, aerosol particles influence the Earth's radiation balance due to their direct extinction of light and their capability to serve as cloud condensation nuclei (CCN) or ice nuclei (IN). These influences result in very high uncertainties in predicting the ongoing climate change (IPCC, 2013)".

*9. (Page 1, lines 20), in terms of number concentrations*
**Response:** "in terms of number concentrations" was added in the manuscript.

*10. (Page 1, lines 24), rephrase "gaseous vapors nucleation"*
**Response:** It was revised to "cluster formation from gaseous vapors gaseous vapors".

*11. (Page 1, lines 29), HOMs instead of HOM*
**Response:** In this study, we recommend to use the term HOM and not HOMs after careful consideration and it is defined as Highly-oxygenated Organic Molecules. All HOMs were revised to HOM accordingly.

*12. (Page 2, lines 25), revise "recentZu years"*
**Response:** Sorry for the spelling mistake. It was revised to "recent years".

*13. (Page 2, lines 28), rephrase "which was later than that observed for SO2"*
**Response:** It was revised to "which appeared later than $SO_2$".

*14. (Page 3, lines 12), revise "including in"*
**Response:** It was revised to "Atmospheric NPF has been observed globally in almost all kinds of environments".

*15. (Page 3, lines 17), The authors stated here that they summarized NPF studies in polluted regions in China. In fact, many included studies are from clean atmospheres, even compared to European countries*
**Response:** It was revised to "In this study, we will summarize the NPF studies conducted in China, focusing on the properties of the NPF events in polluted regions and trying to figure out the possible reasons for the frequent occurrence of NPF at high aerosol loadings".

*16. (Page 3, lines 26), provide evidence that Xi'an and Urumqi are more polluted. Is this only from one study or always true? In fact, a yearly average might be more convincing.*

**Response:** Thanks for the suggestion. We added the yearly average fine particle concentration in 2017 in the revised manuscript. "In 2017, according to the reports of Xi'an Environmental Protection Bureau (http://xaepb.xa.gov.cn/ptl/def/def/index_982_4434_ci_trid_2861812.html, last access: 12 October 2018), Xinjiang Department of Environmental Protection (http://www.xjepb.gov.cn/xjepb/resource/cms/article/2012/268650/2017.pdf, last access: 12 October 2018) and Beijing Municipal Environmental Protection Bureau (http://www.bjepb.gov.cn/bjhrb/resource/cms/2018/05/2018051614522475279.pdf, last access: 12 October 2018), the annual average $PM_{2.5}$ concentration were 73 µg/m$^3$ and 70 µg/m$^3$ in Xi'an and Urumqi, respectively, which were higher than that of Beijing (58 µg/m$^3$)." was added in page 4 lines 2-8.

*17. (Page 3, lines 29-), provide the detailed locations of the sites mentioned, since readers may not be familiar with these Chinese supersites. Also the same for Table 2.*

**Response:** The type of the site and the area where the site is located are introduced in Table 2 and Figure 1, and we think that this information will be enough for a review paper aiming to draw a general picture of NPF observations in the whole country. If the readers are interested in the NPF in one specific observation site, we would suggest them to read the reference directly.

*18. (Session 3.1), In addition to the many factors that have been discussed in the manuscript, emission is a major player in determining the NPF frequency, which is not quite emphasized in the current manuscript.*

**Response:** As mentioned above, the following discussion about the effect of emission on determining NPF frequency was added in the revised manuscript in Page 5 line 26- page 6 line 12:

"A challenge that exists for identifying NPF is the interference of primarily emitted particles from local combustion sources near the observation site. For example, the formation and rapid growth of vehicular particles during the initial 1–2 s of exhaust cooling and dilution processes frequently lead to a nucleation mode at 10–20 nm (Vu et al., 2015;Lee et al., 2015). Spikes of particle number concentration associated with combustion emissions were observed in many NPF studies (Liu et al., 2014;Wang et al., 2014;Peng et al., 2017;Zhu et al., 2017), but these spikes usually had some different characteristics from those of the NPF events (Wang et al., 2014). The particle size (Hofman et al., 2016), the ratio of particle number concentrations in the nucleation mode particles to those of fine particles (Peng et al., 2017;Jung et al., 2013), the time of duration of NPF events (Zhu et al., 2017), and the correlation of the particle number concentration with other gaseous pollutants concentrations and meteorology conditions (Wang et al., 2014) were used to identify the contribution of primary emission in the burst of particle number concentration. However, there are still uncertainties in distinguishing the new-particle signal from the mixed signals of newly-formed particles and freshly-emitted particles from combustion, especially when NPF measurements were carried out with a particle size detection limit larger than 10 nm. There is a possibility that the growth of the vehicular emission of sub-10 nm particles may look like a NPF event and therefore overestimate the NPF frequency. A recently observation found

a notable presence of traffic-originated nanocluster aerosol particles in the size range of 1.3–3.0 nm in urban air (Rönkkö et al., 2017), which might raise new questions about the sources of nanocluster aerosol particles in semi-urban roadside environments."

*19. (Session 3.2), discuss Kerminen-Kulmala equation under the umbrella of Kulmala et al. Faraday Discussions 2018. Deriving J1 from particle formation rate at a larger size may not be feasible.*

**Response:** We agree with the reviewer that the Kerminen-Kulmala equation may not be feasible for deriving nucleation rates from formation rates of larger size particles because this equation requires making several assumptions that may not be valid. This is also one reason that we did not derive $J_1$ from observed particle formation rates at larger particle sizes. Some additional discussion for this issue, i.e. "Besides, the assumed coagulation sticking probability of 1 for molecular clusters with pre-existing particles in their collision and the unclear GR of sub-3 nm particles might result in errors in the derivation of FR (Kulmala et al., 2017)" was added in the revised manuscript in page 8 line 30-page 9 line 2.

*20. (Page 8, lines 19), rephrase "in the NPF Chinese megacities"*

**Response:** It was revised to "in the NPF in Chinese megacities".

*21. (Page 9, lines 15-19), rephrase the sentence*

**Response:** It was revised to "They further calculated the relative contribution of sulfuric acid to the particle growth in different particle size ranges. These calculated contributions were 39% and 29% for the size ranges of 2.39-7 and 7-20 nm, respectively, in urban Shanghai (Xiao et al., 2015), 3% to 14% for the size range of 7-30 nm in urban Beijing (Wang et al., 2015c), about 26 % for the size range of 5.5-25 nm in suburban Hong Kong (Guo et al., 2012), and about 29% during the Beijing Summer Olympic period (Gao et al., 2012)".

*22. (Page 9, lines 32), Tao et al. (2016) looked at really large particles, the composition of which may not be directly related to NPF*

**Response:** Thanks for this comment. It was revised to "Tao et al. (2016) observed higher levels of aminiums in particles with relative smaller sizes, and suggested that the heterogeneous uptake of amines by acid-base reactions could effectively contribute to the particle growth during NPF events. However, they only measured the particle chemical composition with a lowest cutoff size of 56 nm, which may not be directly related to NPF. In fact, measuring the chemical composition of nucleation-mode particles is still quite challenging all over the world." in the revised manuscript.

*23. (Figure 2), assign the references to each of the bars, either in the figure or a number in the figure and details in the figure caption*

**Response:** It was revised accordingly in Figure 2.

**Response for Reviewer #2**

*The authors summarized the data of NPF measurements reported in China in the past 14 years. The features of NPF parameters such as frequency, FR and GR were described, and the possible reasons behind them were also discussed. The author raised attention on the ambiguous relationship between NPF and haze formation, and pointed out the importance of comprehensive study including laboratory works and field measurement on NPF research in China. The data set of this paper is full accurate, could be helpful for other researchers. However, there are still some concerns to be satisfied before I suggest the paper could be published on ACP.*

*Major suggestions: In Section 3, the authors made so far most complete and detailed summary of NPF field measurement results in China. However, on scope of scientific issues, similar understandings were already raised by the review papers from Wang et al, Kulmala et al and Kerminen et al. The authors should strengthen the highlight of this paper by:*

*1. Emphasizing the new findings from studies published in recent years, utilizing the up-to-date techniques, e.g. measurements of sub-3 nm particles and sulfuric acid, since these were not included in former papers.*

**Response:** Thanks for the reviewer's comments. The new findings from studies published in recent years were *emphasized* in the revised manuscript, and the utilizing the up-to-date techniques in recent years in the NPF study in China was also summarized. Some examples of these related revisions are listed here.

Page 4 lines 29-32: "In recent years, an increasing number of studies were carried out with measurements of sub-3 nanoparticles (Xiao et al., 2015;Cai et al., 2017;Cai and Jiang, 2017;Jayaratne et al., 2017;Dai et al., 2017;Lv et al., 2018;Yao et al., 2018), using Particle Size Magnifiers (PSM), neutral cluster and air ion spectrometers (NAIS) or diethylene glycol scanning mobility particle spectrometers (DEG-SMPS)."

Page 5 lines 3-8: "Amines and ammonia are crucial in NPF since they are able to stabilize sulfuric acid clusters by forming acid-base complexes, yet there are very little NPF measurement results related to these compounds in China (Zheng et al., 2015;Yao et al., 2016;Yao et al., 2018). Measurement results on natural ions and neutral compounds/clusters, including both $H_2SO_4$ and HOM, obtained using an atmospheric pressure interface time-of-flight mass spectrometer (APi-TOF-MS) and a nitrate-based chemical ionization–APi-TOF-MS (CI-APi-TOF-MS), were recently reported by Yao et al. (2018)."

Page 7 lines19-20: "higher $SO_2$ concentrations could increase the probability of occurrence of NPF events at a mountaintop site (Lv et al., 2018)."

Page 9 line 27- Page 10 line 4: "According to the national ammonia observation network, the overall average concentration of ammonia in China is much higher than the values observed in the U.S. The seasonal maximum $NH_3$ concentrations were observed in the summer and the most abundant concentrations of $NH_3$ were observed in the NCP region

in China (Pan et al., 2018). Compare to NH$_3$, the amine measurements are more sparse (Zheng et al., 2015b;Yao et al., 2016), and direct information on amine emissions is currently not available but these emissions have to be estimated by assuming a fixed ratio or source-dependent ratios of amines to total ammonia emissions in China (Mao et al., 2018). Dai et al.(2017) proposed that plumes containing high concentrations of ammonia, amines or HOM produced from their observed VOCs led to strong local NPF events. The observations made at the SORPES station in YRD indicated that HOM played an essential role in the initial condensational growth of newly formed clusters (Huang et al., 2016;Ding et al., 2016;Qi et al., 2018)."

Page 11 lines 17-19: "Although the existence of local maxima in GR in the sub-3 nm size range is highly sensitive to uncertainties in particle size distributions, the results highlighted that detailed investigations for the mechanisms of the initial growth steps of atmospheric NPF are needed (Yu et al., 2016)."

Page 16-lines 1-9: "In recent years, an increasing number of studies utilized more advanced instruments, such as PSM (Xiao et al., 2015;Dai et al., 2017;Yu et al., 2016;Yao et al., 2018), NAIS (Jayaratne et al., 2017;Lv et al., 2018), DEG-SMPS (Cai and Jiang, 2017;Cai et al., 2017) and APi-ToF-MS/CI-APi-ToF-MS (Yao et al., 2018), greatly improving our understanding about the nucleation and particle growth mechanisms in China, especially in highly-polluted environments. However, the lack of continuous and comprehensive long-term observations, which should include measurements of particle number size distribution preferably down to 1–2 nm and vapors that potentially participate in NPF and subsequent particle growth (H$_2$SO$_4$, ELVOCs, LVOC, ammonia and amines), still limits our understanding on the mechanism of NPF in different environments in China."

*2. Enriching the discussion in relationship between haze and NPF in Section 4, concentrating on issues such as : 1) threshold of the occurrence of NPF; 2) contribution of NPF on haze formation, in comparison with primary emissions.*

**Response:** We rearranged the consequence of some paragraphs and added some additional discussion. For the threshold of the occurrence of NPF, some discussion was added in the revised manuscript:

Page 12 lines 25-27: "The ratio of particle scavenging loss rate over condensational growth rate, which is proportional to the ratio of CS to GR, was used as a criterion to predict the occurrence of NPF events (McMurry et al., 2005;Kuang et al., 2010)."

Page 13 lines 3-6: "There are several possible reasons for the higher threshold ratio of CS to GR in highly polluted environment, including the overestimation of particle losses due to assuming a coagulation sticking probability of 1, the underestimation of GR in the sub-3 size range, and also unrevealed nucleation and growth mechanism relevant to a polluted atmosphere (Kulmala et al., 2017;Yu et al., 2017)."

For the contribution of NPF to the haze formation, we moved two paragraphs in Section 5 to Section 4, including the effects of NPF to particle number, surface area and mass concentrations in different modes. Relevant publications were summarized in these discussions to analyze the possible contribution of NPF to haze according to different observation results in China. We also added some additional discussion into Page 14 lines 25-30: "In summary, NPF was found to be the main source of the particle number concentration in the atmosphere, being able to dramatically increase particle number

concentrations in a relatively short time. NPF and subsequent particle growth seem also to have a noticeable contribution to the volume and mass concentration of nucleation- and Aitken-mode particles. Although secondary formation of $PM_{2.5}$ mass is the main cause of haze compared with primary particle emissions, the accumulation of this secondary aerosol mass usually occurs over several days following NPF. The contribution of NPF to haze formation is still an open question."

*Minor suggestions:*

*1. Page2 lines 25 "recentZu years" Type error.*

**Response:** Sorry for the type error. It was revised to "recent years".

*2. Page3 lines 26 The air pollution level decreases from NCP to PRD, or from north to south, among these three regions. References are needed.*

**Response:** We added the work of Zhang, Y.-L., and Cao, F., in which one-year (2013-2014) PM2.5 concentrations in 190 cities in China were summarized.

Reference: Zhang, Y.-L., and Cao, F.: Fine particulate matter (PM2.5) in China at a city level, *Scientific Reports*, 5, 10.1038/srep14884, 2015.

*3. Page5 About regional occurrence of NPF, Shen et al's work is encouraged to be cited.*

*Shen, X. J., Sun, J. Y., Kivekas, N., Kristensson, A., Zhang, X. Y., Zhang, Y. M., Zhang, L., Fan, R. X., Qi, X. F., Ma, Q. L., and Zhou, H. G.: Spatial distribution and occurrence probability of regional new particle formation events in eastern China, Atmospheric Chemistry And Physics, 18, 587-599, 10.5194/acp-18-587-2018, 2018.*

**Response:** Thanks for the reminding. This work was added into Page 6 lines 20-21: "Shen et al. (2018) observed regional NPF in NCP with a horizontal extent larger than 500 km and found that large-scale regional NPF was favored by a fast transport of northwesterly air masses."

*4. Page6 Season and the origin of air mass make impact on occurrence of NPF by influencing the level of CS and precursors. The suggestion is moving the discussions on them after those on CS and gaseous pollutants.*

**Response:** This was revised accordingly.

*5. Page7 CS/GR may not be a good index on the probability of NPF, because GR couldn't represent the level of nucleation precursor. The compounds that are crucial for nucleation (e.g. H2SO4, amine) may be negligible in particle growth. The suggestion is providing a review the value of SO2\*UVB/CS in references, since the measurement of H2SO4 was quite limited in China.*

**Response:** Thanks for the reviewer's comments. McMurry et al. (2005) proposed a criterion *L* for new particle formation in the sulfur-rich environment. This criterion is proportional to the ratio of $A_{Fuchs}$ (proportional to CS) and the concentration of

the sulfuric acid ($N_1$). Kuang et al (2010) improved this criterion to $L_\Gamma$ since the old $L$ criterion underestimates the frequency of NPF due to multi-component nucleation/growth and the different species responsible for growth and nucleation. With the assumption that particles throughout the nucleation mode undergo the same enhancement to growth, the growth enhancement factor $\Gamma$, which is the ratio of the measured growth rate to the growth rate assuming free-molecule condensation of sulfuric acid, was used to as a multiplier of the sulfuric acid concentration. CS/GR (proportional to $L_\Gamma$) is proportional to $L$ if that the growth rates are determined by rates of sulfuric acid condensation, i.e. the growth enhancement factor $\Gamma$ is equal to 1 (McMurry et al., 2005;Kuang et al., 2010). We agree that sulfuric acid measurements are quite limited in China, and are measurements of GR for very small particles (such as <3 nm). We tried to review the values of $SO_2$*UVB/CS, but it turned out that very rare references reported these three values. As mentioned earlier, some discussion about the threshold of the occurrence of NPF were added into the revised manuscript.

**References**

[revised manuscript text omitted]